# Effective Dimension Adaptive Sketching Methods for Faster Regularized Least-Squares Optimization

**Jonathan Lacotte**
Department of Electrical Engineering
Stanford University
lacotte@stanford.edu

**Mert Pilanci**
Department of Electrical Engineering
Stanford University
pilanci@stanford.edu

## Abstract

We propose a new randomized algorithm for solving L2-regularized least-squares problems based on sketching. We consider two of the most popular random embeddings, namely, Gaussian embeddings and the Subsampled Randomized Hadamard Transform (SRHT). While current randomized solvers for least-squares optimization prescribe an embedding dimension at least greater than the data dimension, we show that the embedding dimension can be reduced to the effective dimension of the optimization problem, and still preserve high-probability convergence guarantees. In this regard, we derive sharp matrix deviation inequalities over ellipsoids for both Gaussian and SRHT embeddings. Specifically, we improve on the constant of a classical Gaussian concentration bound whereas, for SRHT embeddings, our deviation inequality involves a novel technical approach. Leveraging these bounds, we are able to design a practical and adaptive algorithm which does not require to know the effective dimension beforehand. Our method starts with an initial embedding dimension equal to 1 and, over iterations, increases the embedding dimension up to the effective one at most. Hence, our algorithm improves the state-of-the-art computational complexity for solving regularized least-squares problems. Further, we show numerically that it outperforms standard iterative solvers such as the conjugate gradient method and its pre-conditioned version on several standard machine learning datasets.

## 1 Introduction

We study the performance of a randomized method, namely, the Hessian sketch [34], in the context of regularized least-squares problems,

$$x^* := \underset{x \in \mathbb{R}^d}{\operatorname{argmin}} \left\{ f(x) := \frac{1}{2}\|Ax - b\|_2^2 + \frac{\nu^2}{2}\|x\|_2^2 \right\}, \tag{1}$$

where $A \in \mathbb{R}^{n \times d}$ is a data matrix and $b \in \mathbb{R}^n$ is a vector of observations. For clarity purposes and without loss of generality (by considering instead the dual problem of (1)), we make the assumption that the problem is over-determined, i.e., $n \geqslant d$ and that $\operatorname{rank}(A) = d$.

The regularized solution $x^*$ can be obtained using direct methods which have computational complexity $\mathcal{O}(nd^2)$. In the large-scale setting $n, d \gg 1$, this is prohibitively large. A linear dependence $\widetilde{\mathcal{O}}(nd)$ is preferable and this can be obtained by using first-order iterative solvers [18] such as the conjugate gradient method (CG) for which the per-iteration complexity scales as $\mathcal{O}(nd)$. Using the standard prediction (semi)-norm error $\frac{1}{2}\|\overline{A}(\widetilde{x} - x^*)\|_2^2$ where $\overline{A} := \begin{bmatrix} A \\ \nu I_d \end{bmatrix}$ as the evaluation criterion for an estimator $\widetilde{x}$, these iterative methods have time complexity which usually scales proportionally

to the condition number $\kappa$ of $\overline{A}$ (or $\sqrt{\kappa}$ with acceleration) in order to find a solution $\widetilde{x}$ with acceptable accuracy. This also becomes prohibitively large when $\kappa \gg 1$. Besides the computational complexity, the number of iterations of an iterative solver is also a relevant performance metric in the large-scale setting, as distributed computation may be necessary at each iteration. In this regard, randomized preconditioning methods [37, 4, 29] involve using a random matrix $S \in \mathbb{R}^{m \times n}$ with $m \ll n$ to project the data $A$, and then improve the condition number of $A$ based on a spectral decomposition of $SA$. On the other hand, the iterative Hessian sketch (IHS) introduced by [34] and considered in [30, 25, 26, 31, 35] addresses the conditioning issue differently. Given $x_0, x_1 \in \mathbb{R}^d$, it uses a pre-conditioned Heavy-ball update with step size $\mu$ and momentum parameter $\beta$, given by

$$x_{t+1} = x_t - \mu H_S^{-1} \nabla f(x_t) + \beta(x_t - x_{t-1}) \tag{2}$$

where the Hessian $H := \overline{A}^\top \overline{A}$ of $f(x)$ is approximated by $H_{\overline{S}} = \overline{A}^\top \overline{S}^\top \overline{S} \, \overline{A}$ and $\overline{S}$ is a sketching matrix. We refer to the update (2) as the *Polyak-IHS method*, and, in the absence of acceleration ($\beta = 0$), we call it the *gradient-IHS method*. In contrast to preconditioning methods [37, 4, 29], the IHS does not need to pay the full cost $\mathcal{O}(md \min\{m, d\})$ for decomposing the matrix $SA$. Although solving exactly the linear system $H_{\overline{S}} \cdot z = \nabla f(x_t)$ also takes time $\mathcal{O}(md \min\{m, d\})$, approximate solving (using for instance CG) is also efficient and faster in practice [31, 30].

The choice of the sketching matrix $S$ is critical for statistical and computational performances. A classical sketch is a matrix $S$ with independent and identically distributed (i.i.d.) Gaussian entries $\mathcal{N}(0, m^{-1})$ for which forming $SA$ requires in general $\mathcal{O}(mnd)$ basic operations (using classical matrix multiplication). On the other hand, it has been observed [27, 16] and also formally proved [15, 26] in several contexts that random projections with i.i.d. entries degrade the performance of the approximate solution compared to orthogonal projections. In this regard, the SRHT [1] is an orthogonal embedding for which the sketch $SA$ can be formed in $\mathcal{O}(nd \log m)$ time, and this is much faster than Gaussian projections. Consequently, along with the statistical benefits of orthogonal projections, this suggests to use the SRHT as a reference point for comparing sketching algorithms.

In the context of *unregularized* least-squares problems ($\nu = 0$), [25] showed that the error $\frac{1}{2}\|A(x_t - x^*)\|_2^2$ of the Polyak-IHS method is smaller than $(d/m)^t$ for both Gaussian and SRHT matrices provided that $m \approx d$. More recently, it has been shown in [26] that the scaling $(d/m)^t$ is exact for Gaussian embeddings in the asymptotic regime where we let the relevant dimensions go to infinity, whereas the exact scaling for the SRHT is slightly smaller than $(d/m)^t$.

In the *regularized* case ($\nu > 0$), more relevant than the matrix rank is the *effective dimension* $d_e := \operatorname{trace}(A(A^\top A + \nu^2 I_d)^{-1} A^\top)$ which always satisfies $d_e \leqslant d$, and it is significantly smaller than $d$ when the matrix $A$ has a fast spectral decay. It has been shown in [31] that one can pick $m \approx d_e$ and achieve the error rate $(d_e/m)^t$ by using the well-structured approximate Hessian

$$H_S := A^\top S^\top S A + \nu^2 \cdot I. \tag{3}$$

Further, with $m \approx d_e$ instead of $m \approx d$, the linear system $H_S \cdot z = \nabla f(x_t)$ can be solved in time $\mathcal{O}(d_e^2 d)$ instead of $\mathcal{O}(d^3)$ by computing and caching a factorization of $SA$ and then using the Woodbury matrix identity [20] to invert $H_S$.

However, it is necessary to estimate $d_e$ (which is usually unknown) to be able to pick $m \approx d_e$ and then achieve these computational and memory space savings. The randomized technique proposed by [3] can be used to estimate $d_e$, but under the restrictive assumption that $d_e$ is very small (e.g., see Theorem 60 in [3]). In [31], the authors propose to use a heuristic Hutchinson-type trace estimator [5] and do not provide any guarantee on the estimation accuracy of $d_e$. Consequently, our *main goal* in this paper is to design an adaptive algorithm which does not require the knowledge of $d_e$, but is still able to use a sketch size $m \lesssim d_e$ and achieve an error rate $(d_e/m)^t$.

State-of-the-art randomized preconditioning methods [37, 4, 29] prescribe to use $m$ proportional to $d$ in the context of unregularized least-squares problems. Since it appears non-trivial to adapt and analyze these methods to the regularized case with sketch sizes $m \approx d_e$, nor to design an adaptive scheme which does not require the knowledge of $d_e$, we focus our attention to the Polyak-IHS method in this work.

## 1.1 Notations

We denote by $\|z\|$ or $\|z\|_2$ the Euclidean norm of a vector $z$, $\|M\|_2$ the operator norm of a matrix $M$ and $\|M\|_F$ its Frobenius norm.

We introduce the diagonal matrix $D := \mathrm{diag}\left(\frac{\sigma_1}{\sqrt{\sigma_1^2 + \nu^2}}, \ldots, \frac{\sigma_d}{\sqrt{\sigma_d^2 + \nu^2}}\right)$ where $\sigma_1 \geqslant \ldots \geqslant \sigma_d$ are the singular values of the matrix $A$. We define the effective dimension as $d_e := \frac{\|D\|_F^2}{\|D\|_2^2}$. We denote by $U \in \mathbb{R}^{n \times d}$ a matrix of left singular vectors of $A$ and by $\overline{U} \in \mathbb{R}^{(n+d) \times d}$ a matrix of left singular vectors of $\overline{A} := \begin{bmatrix} A \\ \nu \cdot I_d \end{bmatrix}$.

Given a sequence of iterates $\{x_t\}$, we define its error at time $t$ as $\delta_t := \frac{1}{2}\|\overline{A}(x_t - x^*)\|^2$.

For a sketching matrix $S \in \mathbb{R}^{m \times n}$, we denote the approximate Hessian $H_S := A^\top S^\top S A + \nu^2 I_d$, and the exact Hessian $H := \overline{A}^\top \overline{A}$. Critical to our convergence analysis is the matrix $C_S := D(U^\top S^\top S U - I_d)D + I_d$.

## 1.2 Overview of our contributions

Our main contribution is to propose an iterative method that does not require the knowledge of $d_e$, and is still able to achieve the error rate $\mathcal{O}\left((d_e/m)^t\right)$. Our method is initialized with an arbitrary $m$ (e.g, $m = 1$) and, at each iteration of the Polyak-IHS update (2), it uses an *improvement criterion* to decide whether it should increase $m$ or not. We prove that the *adaptive* sketch size satisfies at each iteration $m \lesssim d_e$ and that our algorithm improves on the state-of-the-art computational complexity for solving regularized least-squares problems.

Our algorithmic parameters and improvement criterion depend on the extreme eigenvalues of $C_S$, and it is then critical for optimal performance to have a sharp estimation of these. For Gaussian embeddings, we provide a sharper constant for well-known Gaussian concentration bounds [24]. Our constant is tight in a worst-case sense, and our analysis is based on a recent extension [39] of Gordon's min-max theorem [17]. In the SRHT case, although similar concentration bounds were already obtained (e.g., see Theorem 1 in [13]), we provide a novel technical approach which generalizes the classical results and analysis proposed in [40].

We evaluate numerically our adaptive algorithm on several standard datasets. We consider two settings: (i) the regularization parameter $\nu$ is fixed; (ii) one aims to compute the several solutions along a regularization path. The latter setting is more relevant to many practical applications [43, 22] where estimating a proper regularization parameter is essential. In both cases, we show empirically that our method is faster than the standard conjugate gradient method and one of the state-of-the-art randomized preconditioning methods [37].

Finally, we address the underdetermined case $d \geqslant n$. By considering the dual of (1) which is itself an overdetermined regularized least-squares problem, we show that our adaptive algorithm and theoretical guarantees apply to this setting. We defer the presentation of these results to Appendix A.2.

## 1.3 Other related work

Another class of sketch-and-solve algorithms project both $A$ and $b$, and then computes $\widetilde{x} := \mathrm{argmin}_x \frac{1}{2}\|SAx - Sb\|_2^2 + \frac{\lambda}{2}\|x\|_2^2$ (see e.g. [16, 33, 32, 38, 14, 7, 8]). In [3], the authors showed that for $m \approx d_e/\varepsilon$, the estimate $\widetilde{x}$ satisfies $f(\widetilde{x}) \leqslant (1 + \varepsilon)f(x^*)$. This can result in large $m$ for even medium accuracy, whereas our method yields an $\varepsilon$-approximate solution with $m \approx d_e$ under the mild requirement that the number of iterations $T$ satisfies $T \approx \log(1/\varepsilon)$. Further, the effective dimension can be efficiently estimated only in limited settings (e.g., see Theorem 60 in [3]). Closely related to our work is the iterative method proposed by [11] for solving underdetermined ridge regression problems. It involves a similar approximation of the Hessian $\overline{A}^\top \overline{A}$ by $H_S$, where the sketch size $m$ depends on the effective dimension $d_e$ as opposed to the data dimension $d$. However their proposed method also requires prior knowledge or estimation of $d_e$. Several sketch-and-solve algorithms [10, 44] for ridge regression were not analyzed in terms of $d_e$ but $d$. In the context of kernel ridge regression, it was shown that Nystrom approximations of kernel matrices have performance guarantees for sketch sizes proportional to the effective dimension [6, 2, 12].

Other versions of the IHS have been proposed in the literature, especially in the context of unregularized least-squares. A fundamentally different version uses the same update (2) but with refreshed sketching matrices, i.e., a new matrix $S$ is sampled at each iteration and independently of the previous

ones, and the approximate Hessian $H_S$ is re-computed. Surprisingly, refreshing embeddings does not improve on using a fixed embedding: it results in the same convergence rate in the Gaussian case [25, 26] and in a slower convergence rate in the SRHT case [26].

## 2 Preliminaries

We provide deterministic convergence guarantees for the Polyak- and gradient-IHS methods, and we relate the convergence rates to the extreme eigenvalues of the matrix $C_S$.

Let $S \in \mathbb{R}^{m \times d}$ be any sketching matrix with arbitrary sketch size $m$, and denote by $\gamma_1$ (resp. $\gamma_d$) the largest (resp. smallest) eigenvalue of $C_S$. Since the matrix $D^\top U^\top S^\top S U D$ is positive semi-definite and $\|D\|_2 < 1$, it holds that $C_S$ is positive definite. Given two real numbers $\Lambda > \lambda > 0$, we define the $S$-measurable event $\mathcal{E}_S := \{\lambda \leqslant \gamma_d \leqslant \gamma_1 \leqslant \Lambda\}$. The proofs of the two next results are based on standard analyses of gradient methods [36], and they are deferred to Appendix B.1.

**Theorem 1.** Consider the step size $\mu_{\mathrm{gd}}(\lambda, \Lambda) := 2/(\frac{1}{\lambda} + \frac{1}{\Lambda})$. Then, conditional on $\mathcal{E}_S$, the gradient-IHS method satisfies at each iteration

$$\frac{\delta_{t+1}}{\delta_t} \leqslant c_{\mathrm{gd}}(\lambda, \Lambda), \quad \text{where} \quad c_{\mathrm{gd}}(\lambda, \Lambda) := \left( \frac{\Lambda - \lambda}{\Lambda + \lambda} \right)^2. \tag{4}$$

**Theorem 2.** Consider the step size $\mu_p(\lambda, \Lambda) := 4/(\frac{1}{\sqrt{\lambda}} + \frac{1}{\sqrt{\Lambda}})^2$ and momentum parameter $\beta_p(\lambda, \Lambda) := \left( \frac{\sqrt{\Lambda} - \sqrt{\lambda}}{\sqrt{\Lambda} + \sqrt{\lambda}} \right)^2$. Then, conditional on $\mathcal{E}_S$, the Polyak-IHS satisfies

$$\limsup_{t \to \infty} \left( \frac{\delta_t}{\delta_0} \right)^{\frac{1}{t}} \leqslant c_p(\lambda, \Lambda), \quad \text{where} \quad c_p(\lambda, \Lambda) := \left( \frac{\sqrt{\Lambda} - \sqrt{\lambda}}{\sqrt{\Lambda} + \sqrt{\lambda}} \right)^2. \tag{5}$$

The above rates $c_{\mathrm{gd}}(\lambda, \Lambda)$ and $c_p(\lambda, \Lambda)$ will play a critical role in the design of our adaptive method. For the gradient-IHS method, it should be noted that we are able to monitor the improvement ratio between two consecutive iterates. However, for the Polyak-IHS method, we only obtain an asymptotic guarantee as $t \to +\infty$. This standard result regarding the Heavy-ball method [36] essentially follows from the fact that the iterates obey a non-symmetric linear dynamical system so that, according to Gelfand's formula, the spectral and operator norms of this linear system only coincide asymptotically.

## 3 Sharp convergence rates for Gaussian and SRHT embeddings

According to Theorems 1 and 2, we need sharp estimates of the extreme eigenvalues of $C_S$ in order to pick optimal parameters for the Polyak- and gradient-IHS methods.

### 3.1 The Gaussian case

We provide a concentration bound on the edge eigenvalues $\gamma_1$ and $\gamma_d$ of the matrix $C_S$ in terms of the aspect ratio $\frac{d_e}{m}$. Our analysis is based on a generalized Gordon's Gaussian comparison theorem [17, 39] and it provides sharper constants than existing results. We defer the proof to Appendix C.1.

**Theorem 3.** Let $\rho, \eta > 0$ be some parameters, and $S \in \mathbb{R}^{m \times n}$ be a Gaussian embedding with $m \geqslant \frac{d_e}{\rho}$. Then, it holds with probability at least $1 - 8e^{-m\rho\eta/2}$ that

$$\begin{cases} \gamma_1 \leqslant 1 - \|D\|^2 + \|D\|_2^2 (1 + \sqrt{c_\eta \rho})^2 \\ \gamma_d \geqslant 1 - \|D\|_2^2 + \|D\|_2^2 (1 - \sqrt{c_\eta \rho})^2, \end{cases} \quad \text{provided that} \quad \rho \in (0, 0.18], \, \eta \in (0.01]. \tag{6}$$

where $c_\eta := (1 + 3\sqrt{\eta})^2$.

The lower[1] and upper bounds (6) are respectively increasing and decreasing in $\|D\|_2$, so that one can replace the potentially unknown quantity $\|D\|_2$ by 1 as follows.

**Definition 3.1** (Practical parameters for Gaussian embeddings)**.** Given $\rho \leqslant 0.18$ and $\eta \leqslant 0.01$, we define the bounds $\lambda_{\rho,\eta} := (1 - \sqrt{c_\eta \rho})^2$ and $\Lambda_{\rho,\eta} := (1 + \sqrt{c_\eta \rho})^2$ where $c_\eta = (1 + 3\sqrt{\eta})^2$. We denote the corresponding algorithmic parameters by $\mu_{\mathrm{gd}}(\rho,\eta) := \mu_{\mathrm{gd}}(\lambda_{\rho,\eta}, \Lambda_{\rho,\eta})$, $\mu_{\mathrm{p}}(\rho,\eta) := \mu_{\mathrm{p}}(\lambda_{\rho,\eta}, \Lambda_{\rho,\eta})$ and $\beta_{\mathrm{p}}(\rho,\eta) := \beta_{\mathrm{p}}(\lambda_{\rho,\eta}, \Lambda_{\rho,\eta})$, and the corresponding convergence rates $c_{\mathrm{gd}}(\rho,\eta) := c_{\mathrm{gd}}(\lambda_{\rho,\eta}, \Lambda_{\rho,\eta})$ and $c_{\mathrm{p}}(\rho,\eta) := c_{\mathrm{p}}(\lambda_{\rho,\eta}, \Lambda_{\rho,\eta})$.

According to Theorems 1 and 2, the closer the bounds on $\gamma_1$ and $\gamma_d$ to 1, the faster the convergence rates of the Gradient- and Polyak-IHS updates. Consequently, one needs to pick both $\rho$ and $\eta$ small. However, this trades off, on the one hand, with a larger sketch size $m$ (i.e., higher computational costs) and, on the other hand, with a weaker probabilistic guarantee. For instance, suppose that $\rho \approx 0.1$ and $\eta \approx 0.01$ are fixed. This results in $m \gtrsim 10^3$ to get low failure probability $e^{-m\rho\eta}$. Such a choice of the sketch size is particularly relevant when $d_e/\rho \gtrsim 10^3$, i.e., $d_e \gtrsim 10^2$, and $\min\{n,d\} \gg 10^3$. On the other hand, in the very small $d_e$ regime, it is harder to keep $m$ close to the *target sketch size* $d_e/\rho$. Since our sketching-based method relies on measure concentration phenomena, this should be expected.

**Remark 3.1.** *Letting $d_e, m \to +\infty$ while keeping the aspect ratio $\rho := \frac{d_e}{m}$ fixed and taking $\eta \sim 1/\sqrt{m}$, our bounds* (6) *converge to the respective limits $1 - \|D\|_2^2 + \|D\|_2^2(1 - \sqrt{\rho})^2$ and $1 - \|D\|_2^2 + \|D\|_2^2(1 + \sqrt{\rho})^2$. When $D = \|D\|_2 \cdot I_d$, these limits are exact as they correspond to the edges of the support of the Marchenko-Pastur distribution [28], so that our bounds are tight in a worst-case sense. Further, we have that $\|C_S - I_d\|_2 \leqslant \|D\|_2^2 \left(2\sqrt{\rho} + \rho\right)(1 + 4\,m^{-\frac{1}{4}})$ with probability at least $1 - 8e^{-\frac{\sqrt{m}\rho}{32}}$, whereas standard Gaussian concentration bounds (e.g., see [24]) states that $\|C_S - I_d\|_2 \leqslant \|D\|_2^2 \left(2\sqrt{\rho} + \rho\right)(1 + c_0)$ with high probability for some universal constant $c_0 > 0$. In contrast, our factor $(1 + 4\,m^{-\frac{1}{4}})$ is asymptotically sharper.*

## 3.2 The SRHT case

We provide a concentration bound in terms of the aspect ratio $C(n, d_e) \cdot \frac{d_e \log(d_e)}{m}$ where we introduced the oversampling factor $C(n, d_e) := \frac{16}{3}(1 + \sqrt{\frac{8 \log(d_e n)}{d_e}})^2$. Under the mild requirement $d_e \gtrsim \log(n)$, this factor satisfies $C(n, d_e) = \mathcal{O}(1)$, so that the latter aspect ratio scales as $\frac{d_e \log(d_e)}{m}$.

Our proof generalizes the results and analysis techniques from the work of J. Tropp [40] who treated the specific case $D = I_d$, and it relies on two powerful matrix inequalities, namely, Lieb's and the matrix Bernstein inequalities [41, 42]. We defer it to Appendix C.2. We note that similar concentration bounds were obtained by [13] using different analysis techniques.

**Theorem 4.** Let $\rho \in (0,1)$ and $m \geqslant C(n, d_e) \cdot \frac{d_e \log(d_e)}{\rho}$. Then it holds with probability at least $1 - 9/d_e$ that $\lambda_\rho \leqslant \gamma_d \leqslant \gamma_1 \leqslant \Lambda_\rho$ where $\lambda_\rho := 1 - \|D\|_2^2 \sqrt{\rho}$ and $\Lambda_\rho := 1 + \|D\|_2^2 \sqrt{\rho}$.

As already discussed in the previous section, the operator norm $\|D\|_2$ might be unknown in practice, but one can replace $\|D\|_2$ by 1 as follows.

**Definition 3.2** (Practical parameters for the SRHT)**.** Given $\rho \in (0,1)$, we define the bounds $\lambda_\rho := 1 - \sqrt{\rho}$ and $\Lambda_\rho := 1 + \sqrt{\rho}$. We denote the corresponding algorithmic parameters by $\mu_{\mathrm{gd}}(\rho) := \mu_{\mathrm{gd}}(\lambda_\rho, \Lambda_\rho)$, $\mu_{\mathrm{p}}(\rho) := \mu_{\mathrm{p}}(\lambda_\rho, \Lambda_\rho)$ and $\beta_{\mathrm{p}}(\rho) := \beta_{\mathrm{p}}(\lambda_\rho, \Lambda_\rho)$, and the corresponding convergence rates $c_{\mathrm{gd}}(\rho) := c_{\mathrm{gd}}(\lambda_\rho, \Lambda_\rho)$ and $c_{\mathrm{p}}(\rho) := c_{\mathrm{p}}(\lambda_\rho, \Lambda_\rho)$.

# 4 An adaptive method free of the knowledge of the effective dimension

We propose a novel adaptive method with time-varying sketch size. Our algorithm does not require the knowledge of $d_e$, but still achieves a fast rate of convergence while keeping $m \lesssim d_e$.

Our method is based on monitoring *an approximation of* the improvement ratio $C_t := \frac{\delta_{t+1}}{\delta_t}$. Given a threshold $\overline{C}$, it proceeds as follows. Starting from an arbitrary initial sketch size (say $m = 1$), we compute at time $t$ a gradient-IHS update $x_{t+1}$. If $C_t \lesssim \overline{C}$, then we accept the update $x_{t+1}$. Otherwise, we reject the update $x_{t+1}$, increase the sketch size by a constant factor (say $m \leftarrow 2m$) and re-compute the sketched matrix $SA$. Since only updates with sufficient improvement are accepted, this method achieves a convergence rate smaller than the chosen threshold $\overline{C}$. Importantly, with, for

instance, Gaussian embeddings, according to Theorems 1 and 3, as soon as the sketch size becomes larger than $\Omega(d_e/\overline{C})$ then all the updates are accepted, so that the number of rejected updates $K$ is finite with $K \lesssim \log(d_e/\overline{C})/\log(2)$. However, computing the exact improvement ratio $C_t$ requires the knowledge of $\overline{A}x^*$, and we alleviate this difficulty as described next.

We provide a proxy of the improvement ratio which is especially compatible with the Gradient- and Polyak-IHS updates. We introduce the *approximate* error $r_t := \frac{1}{2}\|C_S^{-\frac{1}{2}}\overline{U}^\top \overline{A}(x_t - x^*)\|^2$, and the *approximate* ratio $c_t := \frac{r_{t+1}}{r_t}$. In the next result, we relate the approximate error vector $r_t$ with a quantity that can be efficiently computed. We defer the proof to Appendix D.1.

**Lemma 1** (Sketched Newton decrement). *It holds that* $r_t = \frac{1}{2}g_t^\top H_S^{-1} g_t$, *where* $g_t := \nabla f(x_t)$.

Since the IHS forms at each iteration the descent direction $H_S^{-1} g_t$, it is fast to additionally compute the sketched Newton decrement[2] $r_t = \frac{1}{2}g_t^\top H_S^{-1} g_t$ and the approximate improvement ratio $c_t = r_{t+1}/r_t$. Consequently, we can efficiently monitor the ratio $c_t$ as opposed to $C_t$ in order to adapt the sketch size. Provided that $c_t$ and $C_t$ are close enough, this would yield the desired performance. We describe our proposed method in Algorithm 1.

---

**Algorithm 1:** Adaptive Polyak-IHS method.

---

**Input :** $A \in \mathbb{R}^{n \times d}$, $b \in \mathbb{R}^n$, $\nu > 0$, initial sketch size $m \geqslant 1$, initial points $x_0, x_1 \in \mathbb{R}^d$, target convergence rates $\overline{c}_{\mathrm{gd}}, \overline{c}_{\mathrm{p}} \in (0,1)$, gradient descent step size $\mu_{\mathrm{gd}}$, Polyak step size $\mu_{\mathrm{p}} \geqslant 0$ and momentum parameter $\beta_{\mathrm{p}} \geqslant 0$

1   Sample $S \in \mathbb{R}^{m \times n}$ and compute $S_A = SA$.
2   Compute $g_1 = \nabla f(x_1)$, $\widetilde{g}_1 = H_S^{-1} g_1$ and $r_1 = \frac{1}{2}g_1^\top \widetilde{g}_1$.
3   **for** $t = 1, 2, \ldots, T-1$ **do**
4      Compute $x_{\mathrm{p}}^+ = x_t - \mu_{\mathrm{p}}\widetilde{g}_t + \beta_{\mathrm{p}}(x_t - x_{t-1})$, $g_p^+ = \nabla f(x_{\mathrm{p}}^+)$, $\widetilde{g}_p^+ = H_S^{-1} g_p^+$, $r_p^+ = \frac{1}{2}g_p^{+\top}\widetilde{g}_p^+$.
5      Compute the Polyak-IHS improvement ratio $c_{\mathrm{p}}^+ = \left(\frac{r_p^+}{r_1}\right)^{\frac{1}{t}}$.
6      **if** $c_p^+ \leqslant \overline{c}_p$ **then**
7         Set $x_{t+1} = x_p^+$, $g_{t+1} = g_p^+$, $\widetilde{g}_{t+1} = \widetilde{g}_p^+$ and $r_{t+1} = r_p^+$.
8      **else**
9         Compute $x_{\mathrm{gd}}^+ = x_t - \mu_{\mathrm{gd}}\widetilde{g}_t$, $g_{\mathrm{gd}}^+ = \nabla f(x_{\mathrm{gd}}^+)$, $\widetilde{g}_{\mathrm{gd}}^+ = H_S^{-1} g_{\mathrm{gd}}^+$ and $r_{\mathrm{gd}}^+ = \frac{1}{2}g_{\mathrm{gd}}^{+\top}\widetilde{g}_{\mathrm{gd}}^+$.
10         Compute the gradient-IHS improvement ratio $c_{\mathrm{gd}}^+ = \frac{r_{\mathrm{gd}}^+}{r_t}$.
11         **if** $c_{gd}^+ \leqslant \overline{c}_{gd}$ **then**
12            Set $x_{t+1} = x_{\mathrm{gd}}^+$, $g_{t+1} = g_{\mathrm{gd}}^+$, $\widetilde{g}_{t+1} = \widetilde{g}_{\mathrm{gd}}^+$ and $r_{t+1} = r_{\mathrm{gd}}^+$.
13         **else**
14            Set $m := 2m$, sample $S \in \mathbb{R}^{m \times n}$ and compute $S_A = S \cdot A$.
15            Set $\widetilde{g}_t := H_S^{-1} g_t$ and return to Step 4.
16         **end**
17      **end**
18   **end**
19   Return $x_T$.

---

Note that Algorithm 1 computes first a Polyak-IHS update. According to Theorem 2, the relative error of the Polyak-IHS update cannot be tightly controlled in finite-time, but only asymptotically as $t \to +\infty$. This makes difficult to provide guarantees using only the Polyak-IHS update based on monitoring an approximate improvement ratio. Therefore, if the Polyak-IHS update fails, Algorithm 1 computes a gradient-IHS update, whose improvement between two successive iterates can be tightly controlled according to Theorem 1. Hence, Algorithm 1 may compute both updates in order to benefit either from the acceleration of the latter or from the hard convergence guarantees of the former. If both updates do not make enough progress then the sketch size is increased.

## 4.1 Convergence guarantees

We now state high-probability guarantees on the performance of Algorithm 1. We show that the sketch size and the number of rejected steps remain bounded, i.e., $m = \mathcal{O}(d_e/\rho)$ and $K = \mathcal{O}(\log(d_e/\rho))$ for Gaussian embeddings, whereas $m = \mathcal{O}(d_e \log(d_e)/\rho)$ and $K = \mathcal{O}(\log(d_e \log(d_e)/\rho))$ for the SRHT. Further, the convergence rate roughly scales as $\rho^t$. We defer the proofs of the next two results to Appendices B.2 and B.3.

**Theorem 5** (Gaussian embeddings). Let $\rho \leqslant 0.18$ and $\eta \leqslant 0.01$. Suppose that we run Algorithm 1 with $\overline{c}_{\text{gd}} = c_{\text{gd}}(\rho, \eta)$, $\overline{c}_p = c_p(\rho, \eta)$, $\mu_{\text{gd}} = \mu_{\text{gd}}(\rho, \eta)$, $\mu_p = \mu_p(\rho, \eta)$ and $\beta_p = \beta_p(\rho, \eta)$ (see Definition 3.1), and, with an initial sketch size $m_{\text{initial}} \geqslant 1$. Then, it holds with probability at least $1 - 8e^{-d_e \eta/2}$ that, across all iterations, the sketch size remains bounded as

$$m \leqslant 2\,c_0 \cdot \frac{d_e}{\rho}\,, \tag{7}$$

where $c_0$ is a numerical constant which satisfies $c_0 \leqslant 5$. Further, the number of rejected updates is upper bounded as

$$K \leqslant \frac{\log\left(\frac{c_0\,d_e}{m_{\text{initial}}\rho}\right)}{\log 2} + 1\,. \tag{8}$$

Moreover, at any fixed iteration $t \geqslant 1$, it holds with probability at least $1 - 8e^{-d_e \eta/2}$ that the relative error satisfies

$$\frac{\delta_t}{\delta_1} \leqslant 9\left(1 + \frac{\sigma_1^2}{\nu^2}\right)\max\left\{1, \frac{d_e}{m_{\text{initial}}}\right\}c_{\text{gd}}(\rho, \eta)^{t-1}\,. \tag{9}$$

**Theorem 6** (SRHT). Fix $\rho \in (0, 1)$. Suppose that we run Algorithm 1 with $\overline{c}_{\text{gd}} = c_{\text{gd}}(\rho)$, $\overline{c}_p = c_p(\rho)$, $\mu_{\text{gd}} = \mu_{\text{gd}}(\rho)$, $\mu_p = \mu_p(\rho)$ and $\beta_p = \beta_p(\rho)$ (see Definition 3.2), and, with an initial sketch size $m_{\text{initial}} \geqslant 1$. Denote $a_\rho := \frac{1+\sqrt{\rho}}{1-\sqrt{\rho}}$. Then, it holds with probability at least $1 - \frac{9}{d_e}$ that, across all iterations, the sketch size remains bounded as

$$m \leqslant 2\,a_\rho C(n, d_e)\frac{d_e \log d_e}{\rho}\,, \tag{10}$$

and the number of rejected updates is upper bounded as

$$K \leqslant \frac{\log\left(a_\rho C(n, d_e)\frac{d_e \log d_e}{m_{\text{initial}}\rho}\right)}{\log 2} + 1\,. \tag{11}$$

Moreover, it holds almost surely that, across all iterations, the relative error satisfies

$$\frac{\delta_t}{\delta_1} \leqslant 2\left(1 + \frac{\sigma_1^2}{\nu^2}\right)c_{\text{gd}}(\rho)^{t-1}\,. \tag{12}$$

The bound (10) on the sketch size is weaker with the SRHT, which requires an additional factor $\log d_e$. This logarithmic oversampling factor was shown to be necessary for other concentration bounds (see, for instance, the discussions in [21, 40]). On the other hand, the bound (9) on the relative error has an additional factor $\frac{d_e}{m_{\text{initial}}}$ when $m_{\text{initial}} \leqslant d_e$ with Gaussian embeddings. According to our proof of Theorem 6, this follows from the orthogonality of the SRHT which causes less distortions than an i.i.d. Gaussian embedding, especially when the embedding dimension is small.

## 4.2 Time and space complexity

We consider here the SRHT for which computing $SA$ is faster than Gaussian projections. We have the following complexity result, whose proof is deferred to Appendix B.4.

**Theorem 7.** Let $\varepsilon \in (0, 1/2)$ be a given precision such that $\varepsilon \leqslant \frac{\nu^2}{\nu^2+\sigma_1^2}$. Under the hypotheses of Theorem 6, it holds with probability at least $1 - \frac{9}{d_e}$ that the number of iterations to reach a solution $x_T$ such that $\delta_T/\delta_1 \leqslant \varepsilon$ satisfies $T = \mathcal{O}(\frac{\log(1/\varepsilon)}{\log(1/\rho)})$. Thus the total time complexity $\mathcal{C}_\varepsilon$ of Algorithm 1 verifies

$$\mathcal{C}_\varepsilon = \mathcal{O}\left(\log(d_e/\rho)\left(nd\log(d_e/\rho) + \frac{d_e^2 \log^2 d_e}{\rho^2}\,d\right) + nd\frac{\log(1/\varepsilon)}{\log(1/\rho)}\right)\,.$$

The time complexity $\mathcal{C}_\varepsilon$ is decomposed into three terms. Sketching the data matrix takes $\mathcal{O}(nd \log(d_e/\rho))$ time. The cost $\mathcal{O}(\frac{d_e^2 \log^2 d_e}{\rho^2} d)$ corresponds to computing a factorization of $H_S$ using the Woodbury identity (see Appendix B.4 for details). These two costs are multiplied by an extra factor $\mathcal{O}(\log(d_e/\rho))$ which is the maximum number of rejected steps. The last term is the per-iteration complexity $\mathcal{O}(nd)$ times the number of iterations $T = \mathcal{O}(\frac{\log(1/\varepsilon)}{\log(1/\rho)})$. In contrast, other state-of-the-art randomized preconditioning methods [37, 4, 29] prescribe the sketch size $m = \frac{d \log d}{\rho}$ and they are also decomposed into three steps: sketching, factoring, and iterating. Sketching with the SRHT also costs $\mathcal{O}(nd \log(d/\rho))$ and the factoring step takes $\mathcal{O}(\frac{d^3 \log^2 d}{\rho^2})$ time. The iteration part costs $\mathcal{O}(nd \frac{\log(1/\varepsilon)}{\log(1/\rho)})$. This yields the total complexity $\mathcal{C}_{\text{other}} = \mathcal{O}(nd \log(d/\rho) + \frac{d^3 \log^2 d}{\rho^2} + nd \frac{\log(1/\varepsilon)}{\log(1/\rho)})$. Thus, even with the extra factor $\log(d_e/\rho)$ due to the rejected steps, our adaptive method improves on the sketching plus factor costs especially when the effective dimension $d_e$ is much smaller than the data dimension $d$ and thus, on the total complexity.

Regarding space complexity, our method requires $\mathcal{O}(d \cdot d_e \log d_e/\rho)$ space to store the sketched matrix $SA$ whereas the other preconditioning methods needs $\mathcal{O}(d^2 \log d/\rho)$. This is a significant improvement when $d_e$ is much smaller than $d$.

**Remark 4.1.** *Our results developed so far are relevant for a dense data matrix A. On the other hand, it is also of great practical interest to develop efficient methods which address the case of sparse data matrices. If the data matrix A has a few non-zero entries, then embeddings for which the computational complexity of forming SA scales as $\mathcal{O}(nnz(A))$ may be more relevant for our adaptive method. Many deviation bounds similar to those we present in Theorems 3 and 4 exist for sparse embeddings (see, for instance, [11, 23, 13]). We leave the analysis of our adaptive method with sparse embeddings to future work.*

## 5 Numerical experiments

We carry out numerical simulations of Algorithm 1 and we compare it to standard iterative solvers, that is, the CG method and the randomized preconditioned CG (pCG) [37]. Numerical simulations were carried out on a 512Gb desktop station and implemented in Python using its standard numerical linear algebra modules[3].

We consider two evaluation criteria: (i) the cumulative time to compute the solutions up to a given precision $\varepsilon > 0$ along an entire regularization path (several values of $\nu$ in decreasing order) and the memory space required by each sketching-based algorithm as measured by the sketch size $m$, and, (ii) the same criteria but for a fixed value of $\nu > 0$.

We present in Figures 1 and 2 results for two standard datasets (see Appendix A.1 for additional experiments): (i) one-vs-all classification of MNIST digits and (ii) one-vs-all classification of CIFAR10 images.

Except for very large values of the regularization parameter $\nu > 0$ for which the regularized least-squares problem (1) is well-conditioned so that the conjugate gradient method is very efficient, we observe that our method is the fastest and requires less memory space than pCG for computing both the solutions of the entire regularization path and for a fixed value of $\nu$. In particular, pCG uses $m = \frac{d}{\rho}$ for Gaussian embeddings and $m = \frac{d \log d}{\rho}$ for the SRHT. Note that, without a priori knowledge or estimation of the effective dimension $d_e$, these are the best statistical lower bounds on the sketch size known for pCG in order to guarantee convergence. Thus pCG is especially slower at the beginning because the factorization cost scales as $\mathcal{O}(d^3)$ and it requires memory space $\mathcal{O}(d^2)$. In contrast, our method starts with $m = 1$ and $m$ does not exceed $\mathcal{O}(d_e/\rho)$ for Gaussian embeddings and $\mathcal{O}(d_e \log d_e/\rho)$ for the SRHT, as predicted by Theorems 5 and 6. Our adaptive sketch size remains sometimes much smaller than these theoretical upper bounds, and we still have a fast rate of convergence.

We observe in practice that, in Algorithm 1, the Polyak-IHS update is often rejected compared to the gradient-IHS update, especially with the SRHT. Therefore, in addition to Algorithm 1, we consider a variant which does not compute the Polyak-IHS update but only the gradient-IHS update. This

variant enjoys exactly the same convergence guarantees as presented in Theorems 5 and 6. Since it computes only a single candidate update, this variant is faster than Algorithm 1 in the case where the Polyak-IHS update is often rejected.

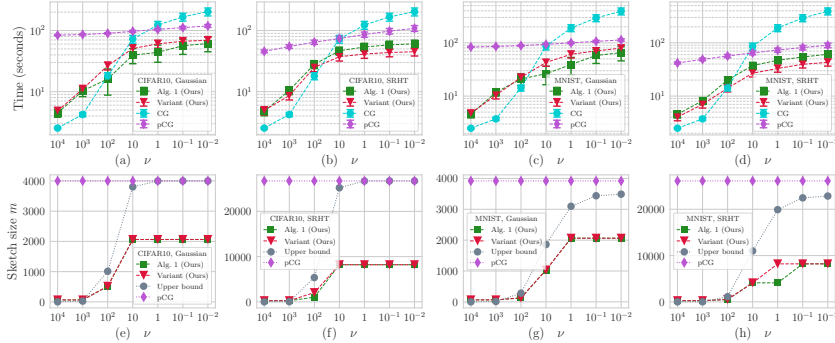

Figure 1: CIFAR10 and MNIST datasets: comparison of CG, pCG, Algorithm 1 and a variant of Algorithm 1 which only computes gradient-IHS updates. We consider an entire regularization path $\nu \in \{10^j \mid j = 4, \ldots, -2\}$. For each algorithm, we start with the largest value $\nu = 10^4$. For $j \leqslant 3$, we initialize each algorithm at the previous solution $\widetilde{x}$ found for $j + 1$. For each value of $\nu$, we stop the algorithm once $\varepsilon = 10^{-10}$-precision is reached. Each run is averaged over 30 independent trials. Mean standard deviations are reported in the form of error bars.

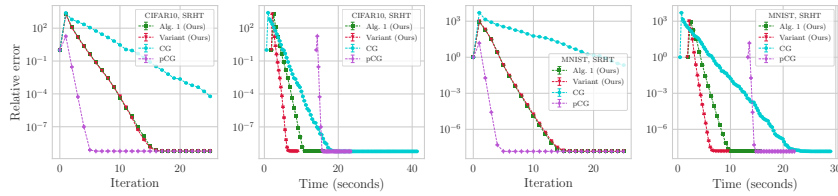

Figure 2: CIFAR10 and MNIST datasets: comparison of CG, pCG, Algorithm 1 and our variant of Algorithm 1 using gradient-IHS updates only. We fix the value of the regularization parameter $\nu = 10$. Each run is averaged over 30 independent trials.

## Broader Impact

We believe that the proposed method in this work can have positive societal impacts. Our algorithm can be applied in massive scale distributed learning and optimization problems encountered in real-life problems. The computational effort can be significantly lowered as a result of adaptive dimension reduction. Consequently energy costs for optimization can be significantly reduced.

## Acknowledgments and Disclosure of Funding

This work was partially supported by the National Science Foundation under grants IIS-1838179 and ECCS-2037304, Facebook Research, Adobe Research and Stanford SystemX Alliance.

## Footnotes

[1]For the lower bound, we use the restrictions $\rho \leqslant 0.18$ and $\eta \leqslant 0.01$ for the sake of having simple expressions, while covering a range of values useful in practice. However, similar lower bounds hold for any $\rho \in (0, 1)$ and small enough $\eta$.

[2]In the optimization literature [9], the Newton decrement at $x$ of a twice differentiable, convex function $f$ is defined as $\frac{1}{2}\nabla f(x)^\top \nabla^2 f(x)^{-1} \nabla f(x)$.

[3]Code is publicly available at https://github.com/jonathanlctt/eff_dim_solver

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
