[Supplementary Material]

# A  Additional results

## A.1  Numerical experiments with synthetic datasets

Here, we consider a synthetic dataset with $A$ having exponential spectral decay $\sigma_j = 0.95^j$ for $j = 1, \ldots, d$. The observation vector is generated as follows, $b = A x_{\mathrm{pl}} + \eta$, where $x_{\mathrm{pl}}$ is a planted vector with $\frac{1}{\sqrt{d}}\mathcal{N}(0,1)$ independent entries and $\eta$ is a vector of Gaussian noise $\frac{1}{\sqrt{n}}\mathcal{N}(0, I_n)$. We also consider the similar synthetic dataset but with polynomially decaying singular values $\sigma_j = 1/j$ for $j = 1, \ldots, d$. Results are reported in Figure 3.

Figure 3: Exponential and polynomial spectral decays: comparison of CG, pCG, Algorithm 1 and a variant of Algorithm 1 which only computes gradient-IHS updates. We consider an entire regularization path $\nu \in \{10^j \mid j = 0, \ldots, -4\}$. For each algorithm, we start with the largest value $\nu = 1$. For $j \leqslant 3$, we initialize each algorithm at the previous solution $\widetilde{x}$ found for $j + 1$. For each value of $\nu$, we stop the algorithm once $\varepsilon = 10^{-10}$-precision is met. We observe that pCG is slow at the beginning due to forming and factoring the $m \times d$ sketched matrix $S \cdot A$ with $m \approx d$. In contrast, our methods start with $m = 1$ and the varying sketch size remains much smaller than that of pCG. This leads to better time and memory space performance, except for the case of Gaussian embeddings and polynomial decays. In the latter case, our method is slowed down by Gaussian projections which are expensive. But with the SRHT, our method has the best performance. Each run is averaged over 30 independent trials. Mean standard deviations are reported in the form of error bars.

## A.2  The underdetermined case $n \leqslant d$

A dual of the problem (1) is

$$ z^* := \operatorname*{argmin}_{z \in \mathbb{R}^n} \left\{ \frac{1}{2}\|A^\top z\|^2 + \frac{\nu^2}{2}\|z\|^2 - b^\top z \right\}, $$

and one can map the optimal dual solution $z^*$ to the primal one using the relationship

$$ x^* = A^\top z^* . \tag{13} $$

The dual problem fits into the primal overdetermined framework we consider in the main body of this manuscript. Indeed, we have that

$$ z^* = \operatorname*{argmin}_{z \in \mathbb{R}^n} \left\{ g(z) := \frac{1}{2}\|A^\top z - \widehat{b}\|^2 + \frac{\nu^2}{2}\|z\|^2 \right\}, \tag{14} $$

where $\widehat{b} = A^\dagger b$ and $A^\dagger$ is the pseudo-inverse of $A$. One might wonder whether $\widehat{b}$ needs to be computed in order to apply the previous framework to the dual overdetermined case: this is not the case. Indeed, in Algorithm 1, the observation vector $b$ only appears in the gradient formula, as $\nabla f(x_t) = A^\top (A x_t - b)$. For the dual problem (14), we have

$$ \nabla g(z_t) = A(A^\top z_t - \widehat{b}) = A A^\top z_t - b . $$

That is, the gradient is easily computed and Algorithm 1 can be applied to the dual problem (14) with the exact same guarantees for the sketch size and the number of rejected steps as in Theorems 5 and 6, while having guarantees on the error

$$\varepsilon_t := \frac{1}{2}\|A^\top(z_t - z^*)\|^2 + \frac{\nu^2}{2}\|z_t - z^*\|^2\,,$$

Using the map $x_t = A^\top z_t$, the notation $\delta_t = \frac{1}{2}\|A(x_t - x^*)\|^2 + \frac{\nu^2}{2}\|x_t - x^*\|^2$ and assuming that $z_0 = 0$ so that $\varepsilon_0 = f(x^*)/\nu^2$, we obtain with Algorithm 1 that $\varepsilon_t \lesssim \rho^t \varepsilon_0$, and consequently

$$\begin{aligned}
\frac{1}{2}\|A(x_t - x^*)\|^2 + \frac{\nu^2}{2}\|x_t - x^*\|^2 &= \frac{1}{2}\|AA^\top(z_t - z^*)\|^2 + \frac{\nu^2}{2}\|A^\top(z_t - z^*)\|^2 \\
&\leqslant \sigma_1(A)^2 \cdot \varepsilon_t \\
&\leqslant \frac{\sigma_1(A)^2 f(x^*)}{\nu^2} \cdot \rho^t\,.
\end{aligned}$$

Thus, the total number of iterations to reach $\varepsilon$-relative accuracy for $x_t$ becomes

$$T = \mathcal{O}\left(\frac{\log(1/\varepsilon) + \log(\sigma_1(A)^2/\nu^2) + \log(f(x^*)/\delta_0))}{\log(1/\rho)}\right)\,.$$

Under the hypothesis $\varepsilon \leqslant \frac{\nu^2}{\nu^2 + \sigma_1(A)^2}$ of Theorem 7 and the additional hypothesis $\frac{f(x^*)}{\delta_0} \leqslant \varepsilon^{-1}$, this number of iterations scales as

$$T = \mathcal{O}\left(\log(1/\varepsilon)/\log(1/\rho)\right)\,.$$

Consequently, we obtain the same total computational complexity (both in time and space) as stated in Theorem 7 to reach an approximate solution $x_t$ with $\varepsilon$-relative accuracy.

## B  Proof of main results

### B.1  Proof of Theorems 1 and 2

We denote by $A = U\Sigma V^\top$ a singular value decomposition of the matrix $A$, where $U = [u_1, \ldots, u_d] \in \mathbb{R}^{n \times d}$ has orthonormal columns, $V = [v_1, \ldots, v_d] \in \mathbb{R}^{d \times d}$ has orthonormal columns, and $\Sigma = \mathrm{diag}(\sigma_1, \ldots, \sigma_d)$, with $\sigma_1 \geqslant \ldots \geqslant \sigma_d > 0$.

We denote $D = \mathrm{diag}\left(\frac{\sigma_1}{\sqrt{\sigma_1^2 + \nu^2}}, \ldots, \frac{\sigma_d}{\sqrt{\sigma_d^2 + \nu^2}}\right)$, $D' = \mathrm{diag}\left(\frac{\nu}{\sqrt{\sigma_1^2 + \nu^2}}, \ldots, \frac{\nu}{\sqrt{\sigma_d^2 + \nu^2}}\right)$, and further,

$$\bar{U} := \begin{bmatrix} UD \\ VD' \end{bmatrix}, \qquad \bar{\Sigma} := \mathrm{diag}\left(\sqrt{\sigma_1^2 + \nu^2}, \ldots, \sqrt{\sigma_d^2 + \nu^2}\right)\,.$$

Note that $\bar{A} = \bar{U}\bar{\Sigma}V^\top$. Indeed,

$$\bar{U}\bar{\Sigma}V^\top = \begin{bmatrix} UD\bar{\Sigma}V^\top \\ VD'\bar{\Sigma}V^\top \end{bmatrix} = \begin{bmatrix} U\Sigma V^\top \\ V(\nu \cdot I_d)V^\top \end{bmatrix} = \begin{bmatrix} A \\ \nu \cdot I_d \end{bmatrix}\,.$$

Further, the columns of $\bar{U}$ are orthonormal, and the matrix $\bar{\Sigma}$ is diagonal with non-negative entries, so that $\bar{U}\bar{\Sigma}V^\top$ is a singular value decomposition of $\bar{A}$.

Given an embedding $S \in \mathbb{R}^{m \times n}$, denote by $\bar{S}$ the $(m+d) \times (n+d)$ block-diagonal matrix $\begin{bmatrix} S & 0 \\ 0 & I_d \end{bmatrix}$.

Denote $\bar{b} = \begin{bmatrix} b \\ 0 \end{bmatrix}$. We have that $\bar{A}^\top \bar{S}^\top \bar{S}\bar{A} = A^\top S^\top SA + \nu^2 I_d = H_S$. Consequently, given a step size $\mu \in \mathbb{R}$ and a momentum parameter $\beta \in \mathbb{R}$, the update formula (2) of the Polyak-IHS method can be equivalently written as

$$x_{t+1} = x_t - \mu(\bar{A}^\top \bar{S}^\top \bar{S}\bar{A})^{-1}\bar{A}^\top(\bar{A}x_t - \bar{b}) + \beta(x_t - x_{t-1})\,. \tag{15}$$

Multiplying the update formula (15) by $\bar{U}^\top \bar{A}$, subtracting $\bar{U}^\top \bar{A}x^*$, using the normal equation $\bar{A}^\top \bar{b} = \bar{A}^\top \bar{A}x^*$ and using the notation $e_t := \bar{U}^\top \bar{A}(x_t - x^*)$, we obtain that

$$
\begin{aligned}
e_{t+1} &= e_t - \mu \bar{U}^\top \bar{A}(\bar{A}^\top \bar{S}^\top \bar{S}\bar{A})^{-1}\bar{A}^\top \bar{U}e_t + \beta(e_t - e_{t-1}) \\
&= \left(I - \mu(\bar{U}^\top \bar{S}^\top \bar{S}\bar{U})^{-1}\right)e_t + \beta(e_t - e_{t-1}).
\end{aligned}
$$

Further, unrolling the expression $\bar{U}^\top \bar{S}^\top \bar{S}\bar{U} = D(U^\top S^\top SU - I_d)D + I_d = C_S$, we find the error recursion

$$
\begin{bmatrix} e_{t+1} \\ e_t \end{bmatrix} = \underbrace{\begin{bmatrix} (1+\beta)I_d - \mu C_S^{-1} & -\beta I_d \\ I_d & 0 \end{bmatrix}}_{:=M(\mu,\beta)} \begin{bmatrix} e_t \\ e_{t-1} \end{bmatrix}. \tag{16}
$$

### B.1.1 Gradient-IHS method

For the gradient-IHS method, we have that $\beta = 0$ so that the dynamics (16) simplifies to

$$
e_{t+1} = (I_d - \mu C_S^{-1})e_t.
$$

Using the fact that $\delta_t = \frac{1}{2}\|e_t\|^2$, we obtain that for any $t \geqslant 0$,

$$
\frac{\delta_{t+1}}{\delta_t} \leqslant \|I_d - \mu C_S^{-1}\|_2^2.
$$

The eigenvalues of the matrix $I_d - \mu C_S^{-1}$ are given by $1 - \frac{\mu}{\gamma_i}$ where the $\gamma_i$'s are the eigenvalues of $C_S$ indexed in non-increasing order. Then,

$$
\|I_d - \mu C_S^{-1}\|_2 = \max\left\{|1 - \frac{\mu}{\gamma_1}|, |1 - \frac{\mu}{\gamma_d}|\right\}.
$$

If $\lambda, \Lambda > 0$ are two real numbers such that $\lambda \leqslant \gamma_d \leqslant \gamma_1 \leqslant \Lambda$, then it holds that for any $\mu \geqslant 0$,

$$
\max\left\{|1 - \frac{\mu}{\gamma_1}|, |1 - \frac{\mu}{\gamma_d}|\right\} \leqslant \max\left\{|1 - \frac{\mu}{\Lambda}|, |1 - \frac{\mu}{\lambda}|\right\}.
$$

Picking $\mu = 2/(\frac{1}{\lambda} + \frac{1}{\Lambda})$ yields that

$$
\|I_d - \mu C_S^{-1}\|_2 \leqslant \left(\frac{\Lambda - \lambda}{\Lambda + \lambda}\right),
$$

which is the result claimed in Theorem 1.

### B.1.2 Polyak-IHS method

Using (16) and the fact that $\delta_t = \frac{1}{2}\|e_t\|^2$, we immediately find by recursion that

$$
\left(\frac{\delta_{t+1} + \delta_t}{\delta_1 + \delta_0}\right)^{\frac{1}{t}} \leqslant \|M(\mu,\beta)^t\|_2^{\frac{2}{t}}.
$$

From Gelfand formula, we obtain that

$$
\limsup_{t\to\infty}\left(\frac{\delta_t}{\delta_0}\right)^{\frac{1}{t}} \leqslant \rho(M(\mu,\beta))^2,
$$

where $\rho(M(\mu,\beta))$ is the spectral radius[4] of the matrix $M(\mu,\beta)$. Let $C_S = T\Lambda T^\top$ be an eigenvalue decomposition of the positive definite matrix $C_S$ – where $\Lambda = \mathrm{diag}(\gamma_1, \ldots, \gamma_d)$ and $\gamma_1 \geqslant \ldots \gamma_d > 0$ –, and define the $(2d) \times (2d)$ permutation matrix $\Pi$ as

$$
\Pi_{i,j} = \begin{cases} 1 & \text{if } i \text{ odd}, \ j = i \\ 1 & \text{if } i \text{ even}, \ j = n + i \\ 0 & \text{otherwise} \end{cases}
$$

Then, it holds that

$$\Pi \begin{bmatrix} T & 0 \\ 0 & T \end{bmatrix}^\top M(\mu,\beta) \begin{bmatrix} T & 0 \\ 0 & T \end{bmatrix} \Pi^\top = \begin{bmatrix} M_1(\mu,\beta) & 0 & \dots & 0 \\ 0 & M_2(\mu,\beta) & \dots & 0 \\ \vdots & & \ddots & \vdots \\ 0 & 0 & \dots & M_d(\mu,\beta) \end{bmatrix}$$

where $M_i(\mu,\beta) = \begin{bmatrix} 1+\beta-\mu\gamma_i^{-1} & -\beta \\ 1 & 0 \end{bmatrix}$. That is, $M(\mu,\beta)$ is similar to the block diagonal matrix with $2 \times 2$ diagonal blocks $M_i(\mu,\beta)$. To compute the eigenvalues of $M(\mu,\beta)$, it suffices to compute the eigenvalues of all of the $M_i(\mu,\beta)$. For fixed $i$, the eigenvalues of the $2 \times 2$ matrix are roots of the equation $u^2 - (1+\beta-\mu/\gamma_i)u + \beta = 0$. In the case that $1 \geqslant \beta \geqslant (1-\sqrt{\mu/\gamma_i})^2$, the roots of the characteristics equations are imaginary, and both have magnitude $\sqrt{\beta}$. Pick $\mu = \mu^*$ : $= 4/(1/\sqrt{\Lambda}+1/\sqrt{\lambda})^2$ and $\beta = \beta^* := \left(\frac{\sqrt{\Lambda}-\sqrt{\lambda}}{\sqrt{\Lambda}+\sqrt{\lambda}}\right)^2$, where $\lambda, \Lambda > 0$ are respectively any lower and upper bounds of $\gamma_d$ and $\gamma_1$. Then, we have that $\beta \geqslant (1-\sqrt{\mu/\gamma_i})^2$ for all $i = 1, \dots, d$, so that $\rho(M(\mu,\beta)) \leqslant \sqrt{\beta}$, and this yields the claimed result. $\qquad\square$

## B.2    Proof of Theorem 5

We introduce the notation $\overline{m} = 5 \cdot \frac{d_e}{\rho}$.

Either the sketch size always remains smaller than $\overline{m}$, which is equivalent to

$$K \leqslant \frac{\log(\overline{m}/m_{\text{initial}})}{\log(2)}, \tag{17}$$

in which case the statements (7) and (8) of Theorem 5 on the sketch size and the number of rejected steps hold almost surely.

Otherwise, suppose that for some iteration $t \geqslant 1$, we have $m > \overline{m}$. Let $\overline{t} \geqslant 1$ be the first such iteration, so that $m \leqslant 2\,\overline{m}$ and $K \leqslant \frac{\log(\overline{m}/m_{\text{initial}})}{\log(2)} + 1$.

Denote $S$ the sketching matrix sampled at time $\overline{t}$. Let $\lambda_{\rho/5,\eta}$ and $\Lambda_{\rho/5,\eta}$ be the bounds as given in Definition 3.1 (where $\rho$ is replaced by $\rho/5$), and consider the event

$$\mathcal{E}_{\rho/5} := \left\{ \lambda_{\rho/5,\eta} \leqslant \sigma_{\min}(C_S) \leqslant \sigma_{\max}(C_S) \leqslant \Lambda_{\rho/5,\eta} \right\}, \tag{18}$$

which, according to Theorem 3 and the fact that $m > \overline{m}$, holds with probability at least $1 - 8e^{-d_e\eta/2}$.

We assume, from now on, that the event $\mathcal{E}_{\rho/5}$ holds. Let $t \geqslant \overline{t}$ be any time such that between $\overline{t}$ and $t$, all updates were accepted (either Polyak- or gradient-IHS), so that the sketch size and sketching matrix are still the same. We claim that *it suffices to prove that the gradient-IHS update at time $t$ is accepted.*

Denote $x_t$ the current iterate, $\delta_t = \frac{1}{2}\|\overline{A}(x_t - x^*)\|^2$ and $r_t = \frac{1}{2}\|C_S^{-\frac{1}{2}}\overline{U}^\top\overline{A}(x_t - x^*)\|^2$. Let $x_{\text{gd}}^+$ be the gradient-IHS update of Algorithm 1, and denote $\delta^+ := \frac{1}{2}\|\overline{A}(x_{\text{gd}}^+ - x^*)\|^2$ and $r^+ := \frac{1}{2}\|C_S^{-\frac{1}{2}}\overline{U}^\top\overline{A}(x_{\text{gd}}^+ - x^*)\|^2$. Recall from Lemma 1 that $r_t$ and $r^+$ are also the sketched Newton decrements at $x_t$ and $x^+$, so that the gradient-IHS improvement ratio computed in Algorithm 1 is equal to $\frac{r^+}{r_t}$.

We need the following technical result whose proof is deferred to Appendix D.2.

**Lemma 2.** Suppose that $\rho \leqslant 0.18$ and $\eta \leqslant 0.01$. Then, on the event $\mathcal{E}_{\rho/5}$, it holds that

$$\frac{\sigma_{\max}(C_S)}{\sigma_{\min}(C_S)} \cdot c_{\text{gd}}(\rho/5,\eta) \leqslant c_{\text{gd}}(\rho,\eta). \tag{19}$$

We have that

$$\frac{\delta^+}{\delta_t} \underset{(i)}{\leqslant} c_{\text{gd}}(\rho/5,\eta) \underset{(ii)}{\leqslant} \frac{\sigma_{\min}(C_S)}{\sigma_{\max}(C_S)} c_{\text{gd}}(\rho,\eta),$$

where inequality (i) follows from Theorem 1, and, inequality (ii) from Lemma 2. Using $r^+ \leqslant \frac{\delta^+}{\sigma_{\min}(C_S)}$ and $r_t \geqslant \frac{\delta_t}{\sigma_{\max}(C_S)}$, it follows that

$$\frac{r^+}{r_t} \leqslant \frac{\sigma_{\max}(C_S)}{\sigma_{\min}(C_S)} \cdot \frac{\delta^+}{\delta_t} \leqslant c_{\mathrm{gd}}(\rho, \eta) \,.$$

Consequently, the gradient-IHS update $x_{\mathrm{gd}}^+$ verifies the improvement criterion $\frac{r^+}{r_t} \leqslant c_{\mathrm{gd}}(\rho, \eta)$, and the update $x_{\mathrm{gd}}^+$ is not rejected.

In summary, as soon as $m > \overline{m}$ and provided that $\mathcal{E}_{\rho/5}$ holds, future updates are not rejected. This holds with probability at least $1 - 8e^{-d_e \eta/2}$, which concludes the proof of the statements (7) and (8) on the sketch size and the number of rejected steps.

We turn to showing statement (9). Fix any iteration $t \geqslant 1$. By construction of Algorithm 1, it holds almost surely that

$$\frac{r_t}{r_1} \leqslant \max\{c_{\mathrm{gd}}(\rho, \eta)^{t-1}, c_{\mathrm{p}}(\rho, \eta)^{t-1}\} = c_{\mathrm{gd}}(\rho, \eta)^{t-1} \,.$$

Denoting by $S$ the sketching matrix at time $t$, and using that $\delta_t \leqslant \sigma_{\max}(C_S) \cdot r_t$ and $\delta_1 \geqslant \sigma_{\min}(C_{S_{\mathrm{initial}}}) \cdot r_1$, it follows that

$$\frac{\delta_t}{\delta_1} \leqslant \frac{\sigma_{\max}(C_S)}{\sigma_{\min}(C_{S_{\mathrm{initial}}})} \cdot \frac{r_t}{r_1} \leqslant \frac{\sigma_{\max}(C_S)}{\sigma_{\min}(C_{S_{\mathrm{initial}}})} \cdot c_{\mathrm{gd}}(\rho, \eta)^{t-1} \,.$$

On the one hand, according to Theorem 3, we have that

$$\sigma_{\max}(C_S) \leqslant \frac{\nu^2}{\sigma_1^2 + \nu^2} + \frac{\sigma_1^2}{\sigma_1^2 + \nu^2} \cdot \left(1 + \sqrt{(1 + 3\sqrt{\eta})^2 \frac{d_e}{m_{\mathrm{initial}}}}\right)^2 \,.$$

with probability at least $1 - 8e^{-\eta d_e/2}$. Using that $\eta \leqslant 0.01$, $(1 + 3\sqrt{\eta}) \leqslant 3/2$ and $(1 + \sqrt{\frac{d_e}{m}})^2 \leqslant 4 \max\{1, \frac{d_e}{m_{\mathrm{initial}}}\}$, we obtain

$$\sigma_{\max}(C_S) \leqslant 9 \left(\frac{\nu^2}{\sigma_1^2 + \nu^2} + \frac{\sigma_1^2}{\sigma_1^2 + \nu^2} \max\{1, \frac{d_e}{m_{\mathrm{initial}}}\}\right) \,.$$

On the other hand, it holds almost surely that

$$\sigma_{\min}(C_{S_{\mathrm{initial}}}) \geqslant 1 - \|D\|_2^2 = \frac{\nu^2}{\sigma_1^2 + \nu^2} \,.$$

Combining the latter inequalities, it holds with probability at least $1 - 8e^{-d_e \eta/2}$ that

$$\frac{\delta_t}{\delta_1} \leqslant 9 \left(1 + \frac{\sigma_1^2}{\nu^2}\right) \max\left\{1, \frac{d_e}{m_{\mathrm{initial}}}\right\} c_{\mathrm{gd}}(\rho, \eta)^{t-1} \,,$$

which concludes the proof.

### B.3 Proof of Theorem 6

The proof for the SRHT follows steps similar to the Gaussian case. We introduce the notation

$$\overline{m} = a_\rho \cdot C(n, d_e) \frac{d_e \log(d_e)}{\rho} \,, \tag{20}$$

and we recall that $a_\rho := \frac{1 + \sqrt{\rho}}{1 - \sqrt{\rho}}$.

Either the sketch size always remains smaller than $\overline{m}$. The latter is equivalent to

$$K \leqslant \frac{\log(\overline{m}/m_{\mathrm{initial}})}{\log(2)} \,, \tag{21}$$

in which case the statements (10) and (11) of Theorem 6 on the sketch size and the number of rejected steps hold almost surely.

Otherwise, suppose that for some iteration $t \geqslant 1$, we have $m > \overline{m}$. Let $\overline{t} \geqslant 1$ be the first such iteration, so that $m \leqslant 2\overline{m}$ and $K \leqslant \frac{\log(\overline{m}/m_{\text{initial}})}{\log(2)} + 1$.

Denote $S$ the sketching matrix sampled at time $\overline{t}$. Define $\lambda_{\rho/a_\rho} := 1 - \sqrt{\frac{\rho}{a_\rho}}$ and $\Lambda_{\rho/a_\rho} := 1 + \sqrt{\frac{\rho}{a_\rho}}$, and consider the event

$$\mathcal{E}_{\rho/a_\rho} := \left\{ \lambda_{\rho/a_\rho} \leqslant \sigma_{\min}(C_S) \leqslant \sigma_{\max}(C_S) \leqslant \Lambda_{\rho/a_\rho} \right\}, \tag{22}$$

which, according to Theorem 4 and the fact that $m > \overline{m}$, holds with probability at least $1 - \frac{d_e}{9}$.

We assume, from now on, that the event $\mathcal{E}_{\rho/a_\rho}$ holds. Let $t \geqslant \overline{t}$ be any time such that between $\overline{t}$ and $t$, all updates were accepted (either Polyak- or gradient-IHS), so that the sketch size and sketching matrix are the same. We claim that *it suffices to prove that the gradient-IHS update at time $t$ is accepted.*

Denote $x_t$ the current iterate, $\delta_t = \frac{1}{2}\|\overline{A}(x_t - x^*)\|^2$ and $r_t = \frac{1}{2}\|C_S^{-\frac{1}{2}}\overline{U}^\top \overline{A}(x_t - x^*)\|^2$. Let $x_{\text{gd}}^+$ be the gradient-IHS update of Algorithm 1, and denote $\delta^+ := \frac{1}{2}\|\overline{A}(x_{\text{gd}}^+ - x^*)\|^2$ and $r^+ := \frac{1}{2}\|C_S^{-\frac{1}{2}}\overline{U}^\top \overline{A}(x_{\text{gd}}^+ - x^*)\|^2$. Recall from Lemma 1 that $r_t$ and $r^+$ are also the sketched Newton decrements at $x_t$ and $x^+$, so that the gradient-IHS improvement ratio computed in Algorithm 1 is equal to $\frac{r^+}{r_t}$.

We need the following technical result whose proof is deferred to Appendix D.3.

**Lemma 3.** On the event $\mathcal{E}_{\rho/a_\rho}$, it holds that $\frac{\sigma_{\max}(C_S)}{\sigma_{\min}(C_S)} \leqslant a_\rho$ and $c_{\text{gd}}(\rho/a_\rho) = \frac{c_{\text{gd}}(\rho)}{a_\rho}$.

We have that

$$\frac{\delta^+}{\delta_t} \underset{(i)}{\leqslant} c_{\text{gd}}(\rho/a_\rho) \underset{(ii)}{=} \frac{c_{\text{gd}}(\rho)}{a_\rho},$$

where inequality (i) follows from Theorem 1, and, equality (ii) from the second part of Lemma 3. Using $r^+ \leqslant \frac{\delta^+}{\sigma_{\min}(C_S)}$ and $r_t \geqslant \frac{\delta_t}{\sigma_{\max}(C_S)}$, it follows that

$$\frac{r^+}{r_t} \leqslant \frac{\sigma_{\max}(C_S)}{\sigma_{\min}(C_S)} \cdot \frac{\delta^+}{\delta_t} \leqslant \frac{\sigma_{\max}(C_S)}{\sigma_{\min}(C_S)} \cdot \frac{c_{\text{gd}}(\rho)}{a_\rho} \underset{(i)}{\leqslant} a_\rho \cdot \frac{c_{\text{gd}}(\rho)}{a_\rho} = c_{\text{gd}}(\rho),$$

where inequality (i) follows from the first part of Lemma 3. Consequently, the gradient-IHS update $x_{\text{gd}}^+$ verifies the improvement criterion $\frac{r^+}{r_t} \leqslant c_{\text{gd}}(\rho)$, and the update $x_{\text{gd}}^+$ is not rejected.

In summary, as soon as $m > \overline{m}$ and provided that $\mathcal{E}_{\rho/a_\rho}$ holds, future updates are not rejected. This holds with probability at least $1 - \frac{9}{d_e}$, which concludes the proof of the statements (10) and (11) on the sketch size and the number of rejected steps.

We turn to showing statement (12). Fix any iteration $t \geqslant 1$. By construction of Algorithm 1, it holds almost surely that

$$\frac{r_t}{r_1} \leqslant \max\{c_{\text{gd}}(\rho)^{t-1}, c_{\text{p}}(\rho)^{t-1}\} = c_{\text{gd}}(\rho)^{t-1}.$$

Denoting by $S$ the sketching matrix at time $t$, and using that $\delta_t \leqslant \sigma_{\max}(C_S) \cdot r_t$ and $\delta_1 \geqslant \sigma_{\min}(C_{S_{\text{initial}}}) \cdot r_1$, it follows that

$$\frac{\delta_t}{\delta_1} \leqslant \frac{\sigma_{\max}(C_S)}{\sigma_{\min}(C_{S_{\text{initial}}})} \cdot \frac{r_t}{r_1} \leqslant \frac{\sigma_{\max}(C_S)}{\sigma_{\min}(C_{S_{\text{initial}}})} \cdot c_{\text{gd}}(\rho)^{t-1}.$$

On the one hand, it holds almost surely that

$$\sigma_{\max}(C_S) = \sup_{\|x\|_2=1} \|x\|_2^2 + \langle Dx, (U^\top S^\top SU - I_d)Dx \rangle$$

$$\underset{(i)}{\leqslant} 1 + \sup_{\|x\|_2 \leqslant 1} \langle x, (U^\top S^\top SU - I_d)x \rangle$$

$$\leqslant 1 + \sup_{\|x\|_2 \leqslant 1} \langle x, U^\top S^\top SUx \rangle$$

$$\underset{(ii)}{\leqslant} 2 \,,$$

where inequality (i) follows from the fact that $\|D\|_2 \leqslant 1$, and inequality (ii) from the fact that $SU$ is a partial orthogonal matrix so that $\|SU\|_2 \leqslant 1$. On the other hand, it holds almost surely that

$$\sigma_{\min}(C_{S_{\text{initial}}}) \geqslant 1 - \|D\|_2^2 = \frac{\nu^2}{\sigma_1^2 + \nu^2} \,.$$

Combining the latter inequalities, it holds almost surely that

$$\frac{\delta_t}{\delta_0} \leqslant 2 \left( 1 + \frac{\sigma_1^2}{\nu^2} \right) c_{\text{gd}}(\rho)^{t-1} \,,$$

which concludes the proof.

$\square$

## B.4  Proof of Theorem 7

According to Theorem 6, we have with probability at least $1 - \frac{9}{d_e}$ that over an entire trajectory, the sketch size and the number of rejected steps satisfy

$$m = \mathcal{O}(d_e \log d_e / \rho) \,, \quad K = \mathcal{O}(\log(d_e/\rho)) \,.$$

From now on, we assume that the above event holds.

Then, forming the sketched matrix $SA$ costs at most $\mathcal{O}(nd \log d_e)$ at any iteration. Using the Woodbury matrix identity, the inverse of $H_S$ verifies

$$H_S^{-1} = \left((SA)^\top SA + \nu^2 I_d\right)^{-1} = \frac{1}{\nu^2} \left( I_d - (SA)^\top (\nu^2 I_m + SA(SA)^\top)^{-1} SA \right) \,.$$

To reduce the complexity of solving at each iteration the linear system $H_S \cdot z = \nabla f(x_t)$, one can simply compute and cache a factorization of the matrix $(\nu^2 I_m + SA(SA)^\top)$ which takes time $\mathcal{O}(\frac{d_e^2 \log^2 d_e}{\rho^2} d)$. Consequently, the total sketching and factor costs scale as $\mathcal{O}(\log(d_e/\rho) \cdot (\frac{d_e^2 \log^2 d_e}{\rho^2} d + nd \log(d_e/\rho)))$.

The per-iteration cost is that of computing the matrix-vector products $Ax_t$ and $A^\top(Ax_t - b)$, which is given by $\mathcal{O}(nd)$. Note that the other main numerical operation consists in solving the linear system $H_S \cdot z = \nabla f(x_t)$. Using the cached factorization of the matrix $(\nu^2 I_m + SA(SA)^\top)$ and the Woodbury identity, this linear system can be solved in time $\mathcal{O}(\frac{d_e \log d_e}{\rho} d)$, which is negligible compared to $\mathcal{O}(nd)$.

According to Theorem 6, we have almost surely that over an entire trajectory,

$$\frac{\delta_{t+1}}{\delta_1} \leqslant 2 \cdot (1 + \frac{\sigma_1^2}{\nu^2}) \cdot c_{\text{gd}}(\rho)^t \,.$$

A simple calculation yields that $c_{\text{gd}}(\rho) = \rho$. Therefore, a sufficient number of iterations $T$ to reach an $\varepsilon$-accurate solution is exactly given by

$$T = \left\lceil \frac{\log 2 + \log(1 + \frac{\sigma_1^2}{\nu^2}) + \log(1/\varepsilon)}{\log(1/\rho)} \right\rceil \,.$$

For $\varepsilon \leqslant \min\{\frac{\nu^2}{\sigma_1^2+\nu^2}, 1/2\}$, this reduces to

$$T = \mathcal{O}\left(\frac{\log(1/\varepsilon)}{\log(1/\rho)}\right).$$

Thus, we obtain the total time complexity

$$C_\varepsilon = \mathcal{O}\left(\log(d_e/\rho) \cdot (\frac{d_e^2 \log^2 d_e}{\rho^2} d + nd\log(d_e/\rho)) + nd \frac{\log(1/\varepsilon)}{\log(1/\rho)}\right),$$

which is the claimed result.

## C  Proofs of concentration inequalities

### C.1  Gaussian concentration over ellipsoids – Proof of Theorem 3

Let $\rho > 0$ and $m \geqslant \frac{d_e}{\rho}$. Let $S \in \mathbb{R}^{m \times n}$ be a random matrix with i.i.d. entries $\mathcal{N}(0, 1/m)$. We aim to control the quantities

$$\gamma_1 = \sup_{\|x\|=1} 1 + \langle x, D(U^\top S^\top SU - I_d)Dx \rangle$$

$$\gamma_d = \inf_{\|x\|=1} 1 + \langle x, D(U^\top S^\top SU - I_d)Dx \rangle.$$

**Upper bound on the largest eigenvalue $\gamma_1$**

We introduce the re-scaled matrix $\bar{D} = \frac{D}{\|D\|_2}$, so that $\|\bar{D}\|_F^2 = d_e$ and $\|\bar{D}\|_2 = 1$. We have that

$$\frac{\gamma_1 - 1}{\|D\|_2^2} \stackrel{\mathrm{d}}{=} \sup_{\|x\|=1} \langle x, \bar{D}(\frac{1}{m}G^\top G - I)\bar{D}x \rangle = \sup_{\|x\|=1} \frac{1}{m}\|G\bar{D}x\|^2 - \|\bar{D}x\|^2$$

$$= \frac{2}{m} \sup_{z \in \mathcal{C}} \sup_{u \in \mathbb{R}^m} u^\top Gz + \psi(u,z),$$

where we introduced the random matrix $G \in \mathbb{R}^{m \times d}$ with i.i.d. Gaussian entries $\mathcal{N}(0,1)$ and the first equality holds since $SU \stackrel{\mathrm{d}}{=} \frac{1}{\sqrt{m}}G$. We also used the notations $\mathcal{C} = \{\bar{D}x \mid \|x\| = 1\}$ and $\psi(u,z) := -\frac{1}{2}(\|u\|^2 + m\|z\|^2)$. We introduce the auxiliary random variable

$$Y := \frac{2}{m} \sup_{z \in \mathcal{C}} \sup_{u \in \mathbb{R}^m} \|z\| g^\top u + \|u\| h^\top z + \psi(u,z),$$

where $g \in \mathbb{R}^m$ and $h \in \mathbb{R}^d$ are random vectors with i.i.d. entries $\mathcal{N}(0,1)$. Using Theorem 9 (see Appendix C.1.1), it holds that for any $c \in \mathbb{R}$,

$$\mathbb{P}\left(\frac{\gamma_1 - 1}{\|D\|_2^2} \geqslant c\right) \leqslant 2\,\mathbb{P}(Y \geqslant c). \tag{23}$$

Consequently, it suffices to control the upper tail of $Y$ in order to control that of $\gamma_1$. First, we recall a few basic facts on the concentration of Gaussian random vectors (see, for instance, Theorems 3.1.1 and 6.3.2 in [42]). That is, for any $\eta > 0$, the following event holds with probability at least $1 - 4e^{-m\rho\eta/2}$,

$$\mathcal{E}_\eta := \left\{ |\|g\| - \sqrt{m}| \leqslant \sqrt{m\eta\rho}, \qquad |\|g\|^2 - m| \leqslant m\sqrt{\eta\rho}, \qquad \|\bar{D}h\| \leqslant \sqrt{m\rho}(1 + \sqrt{\eta}) \right\}, \tag{24}$$

On the event $\mathcal{E}_\eta$, we have

$$Y = \frac{2}{m} \sup_{z \in \mathcal{C}} \sup_{u \in \mathbb{R}^m} \|z\| g^\top u + \|u\| h^\top z - \frac{1}{2}\|u\|^2 - \frac{m}{2}\|z\|^2$$

$$\overset{(i)}{=} \frac{2}{m} \sup_{z \in \mathcal{C}} \sup_{t \geqslant 0} t\, \|z\|\|g\| + t\, h^\top z - \frac{1}{2}t^2 - \frac{m}{2}\|z\|^2$$

$$\overset{(ii)}{\leqslant} \frac{2}{m} \sup_{z \in \mathcal{C}} \sup_{t \in \mathbb{R}} t(\|z\|\|g\| + |h^\top z|) - \frac{1}{2}t^2 - \frac{m}{2}\|z\|^2$$

$$\overset{(iii)}{\leqslant} \frac{2}{m} \sup_{z \in \mathcal{C}} \frac{\|z\|^2}{2}|\|g\|^2 - m| + \frac{1}{2}|h^\top z|^2 + \|z\|\|g\||h^\top z|$$

$$\overset{(iv)}{\leqslant} \frac{2}{m} \sup_{z \in \mathcal{C}} \frac{|\|g\|^2 - m|}{2} + \frac{1}{2}|h^\top z|^2 + \|g\||h^\top z|$$

$$\overset{(v)}{=} \frac{|\|g\|^2 - m|}{m} + \frac{\|\bar{D}h\|^2}{m} + \frac{2\|\bar{D}h\|\|g\|}{m}$$

$$\overset{(vi)}{\leqslant} \sqrt{\rho\eta} + \rho(1 + \sqrt{\eta})^2 + 2\sqrt{\rho}(1 + \sqrt{\eta})(1 + \sqrt{\rho\eta})$$

$$= \rho(1 + \sqrt{\eta})(1 + 3\sqrt{\eta}) + 2\sqrt{\rho}(1 + \frac{3}{2}\sqrt{\eta})$$

$$\leqslant \left(1 + \sqrt{\rho c_\eta}\right)^2 - 1\,,$$

where $c_\eta := (1 + 3\sqrt{\eta})^2$. In equality (i), we used the fact that for a vector $u$ with fixed norm $\|u\| = t$, the maximum of $g^\top u$ is equal to $\|g\|t$. In inequality (ii), we bounded $h^\top z$ by $|h^\top z|$ and then relaxed the constraint $t \geqslant 0$ to $t \in \mathbb{R}$. In inequality (iii), we plugged-in the value of the maximizer $t^* = \|z\|\|g\| + |h^\top z|$. In inequality (iv), we used the fact that for $z \in \mathcal{C}$, $\|z\| \leqslant 1$. In (v), we used the fact that $\sup_{z \in \mathcal{C}} |h^\top z| = \|\bar{D}h\|$. In (vi), we used that, on the event $\mathcal{E}_\eta$, we have $\frac{|\|g\|^2 - m|}{m} \leqslant \sqrt{\eta m}$, $\|\bar{D}h\| \leqslant \sqrt{m\rho}(1 + \sqrt{\eta})$ and $\|g\| \leqslant \sqrt{m}(1 + \sqrt{\eta\rho})$. Consequently, we have that

$$\mathbb{P}\left[\frac{\gamma_1 - 1}{\|D\|_2^2} \geqslant (1 + \sqrt{\rho c_\eta})^2 - 1\right] \leqslant 2\,\mathbb{P}\left[Y \geqslant (1 + \sqrt{\rho c_\eta})^2 - 1\right]$$

$$\leqslant 2(1 - \mathbb{P}[\mathcal{E}_\eta])$$

$$\leqslant 8 \cdot e^{-m\rho\eta/2}\,,$$

which is the claimed upper bound (6) on $\gamma_1$.

**Controlling the smallest eigenvalue $\gamma_d$**

Here we assume that $\rho \in (0, 0.18]$ and $\eta \in (0, 0.01]$. We make this assumption in order to provide explicit and simple statements.

We consider the same definitions $\bar{D}, \mathcal{C}, \varphi$ and $\mathcal{E}_\eta$ introduced in the proof of the upper bound on $\gamma_1$. We have that

$$\frac{\gamma_d - 1}{\|D\|_2^2} \overset{\mathrm{d}}{=} \inf_{\|x\|=1} \langle x, \bar{D}(\frac{1}{m}G^\top G - I)\bar{D}x \rangle = \inf_{\|x\|=1} \frac{1}{m}\|G\bar{D}x\|^2 - \|\bar{D}x\|^2$$

$$= \frac{2}{m} \inf_{z \in \mathcal{C}} \sup_{u \in \mathbb{R}^m} u^\top Gz + \psi(u, z)\,.$$

We introduce the auxiliary random variable

$$Y := \frac{2}{m} \inf_{z \in \mathcal{C}} \sup_{u \in \mathbb{R}^m} \|z\| g^\top u + \|u\| h^\top z + \psi(u, z)\,,$$

where $g \in \mathbb{R}^m$ and $h \in \mathbb{R}^d$ are random vectors with i.i.d. entries $\mathcal{N}(0, 1)$. Using Theorem II.1 from [39], it holds that for any $c \in \mathbb{R}$,

$$\mathbb{P}(\frac{\gamma_d - 1}{\|D\|_2^2} < c) \leqslant 2\,\mathbb{P}(Y < c)\,. \tag{25}$$

Consequently, it suffices to control the lower tail of $Y$ in order to control that of $\gamma_d$. It holds that

$$
\begin{aligned}
Y &= \frac{2}{m} \inf_{z \in \mathcal{C}} \sup_{u \in \mathbb{R}^m} \|z\| g^\top u + \|u\| h^\top z - \frac{1}{2} \|u\|^2 - \frac{m}{2} \|z\|^2 \\
&= \frac{2}{m} \inf_{z \in \mathcal{C}} \sup_{t \geqslant 0} t \|z\| \|g\| + t h^\top z - \frac{1}{2} t^2 - \frac{m}{2} \|z\|^2 \\
&= \inf_{z \in \mathcal{C}} \begin{cases} -\|z\|^2, & \text{if } \|z\| \|g\| + h^\top z \leqslant 0 \\ \frac{\|z\|^2}{m} (\|g\|^2 - m) + \frac{(h^\top z)^2}{m} + \frac{2}{m} \|z\| \|g\| (h^\top z), & \text{otherwise.} \end{cases}
\end{aligned}
$$

Define

$$
Y_1 := \inf_{\substack{z \in \mathcal{C}; \\ \|z\|\|g\| + h^\top z \leqslant 0}} -\|z\|^2,
$$

$$
Y_2 := \inf_{\substack{z \in \mathcal{C} \\ \|z\|\|g\| + h^\top z \geqslant 0}} \frac{\|z\|^2}{m} (\|g\|^2 - m) + \frac{(h^\top z)^2}{m} + \frac{2}{m} \|z\| \|g\| (h^\top z),
$$

so that $Y = \min\{Y_1, Y_2\}$. For any $z \in \mathcal{C}$, it holds that $h^\top z \geqslant -\|\bar D h\|$, and consequently

$$
Y_1 \geqslant \inf_{\substack{z \in \mathcal{C}; \\ \|z\|\|g\| \leqslant \|\bar D h\|}} -\|z\|^2 \geqslant -\frac{\|\bar D h\|^2}{\|g\|^2}.
$$

Hence, conditional on the event $\mathcal{E}_\eta$, we have

$$
Y_1 \geqslant -\rho \left( \frac{1 + \sqrt{\eta}}{1 - \sqrt{\rho \eta}} \right)^2,
$$

On the other hand, we have

$$
\begin{aligned}
Y_2 &\geqslant -\frac{1}{m} \big| \|g\|^2 - m \big| + \inf_{z \in \mathcal{C}} \left\{ \frac{(h^\top z)^2}{m} - \frac{2}{m} \|g\| \|h^\top z\| \right\} \\
&\geqslant -\frac{1}{m} \big| \|g\|^2 - m \big| + \inf_{\|x\| = 1} \left\{ \frac{\langle \bar D h, x \rangle^2}{m} - \frac{2}{m} \|g\| |\langle \bar D h, x \rangle| \right\} \\
&= -\frac{1}{m} \big| \|g\|^2 - m \big| + \frac{2}{m} \inf_{0 \leqslant t \leqslant \|\bar D h\|} \left\{ \frac{t^2}{2} - \|g\| t \right\},
\end{aligned}
$$

where, in the first inequality, we relaxed the constraint set by removing the constraint $\|z\|\|g\| + h^\top z \geqslant 0$ and we used the fact that $\|z\| \leqslant 1$. In the second inequality, we used the change of variable $z = \bar D x$ with $\|x\| = 1$. In the third inequality, we used the fact that $|\langle \bar D h, x \rangle| \leqslant \|\bar D h\|$ and used the change of variable $|\langle \bar D h, x \rangle| = t$ with $t \in [0, \|\bar D h\|]$. On the event $\mathcal{E}_\eta$, it follows that

$$
\begin{aligned}
Y_2 &\geqslant \rho(1 - \eta) - 2\sqrt{\rho}\left(1 + \frac{3}{2}\sqrt{\eta}\right) \\
&\underset{(i)}{\geqslant} (1 + 3\sqrt{\eta})^2 \rho - 2\sqrt{\rho}(1 + 3\sqrt{\eta}) \\
&= (1 - \sqrt{c_\eta \rho})^2 - 1,
\end{aligned}
$$

One can verify that inequality (i) is equivalent to $\sqrt{\rho} \leqslant \frac{1}{2 + \frac{10\sqrt{\eta}}{3}}$, which always holds under the assumption that $\rho \leqslant 0.18$ and $\eta \leqslant 0.01$. Then, combining the respective lower bounds on $Y_1$ and $Y_2$, we obtain that

$$
\begin{aligned}
Y &\geqslant \min \left\{ -\rho \left( \frac{1 + \sqrt{\eta}}{1 - \sqrt{\eta}} \right)^2, (1 - \sqrt{c_\eta \rho})^2 - 1 \right\} \\
&\geqslant (1 - \sqrt{c_\eta \rho})^2 - 1,
\end{aligned}
$$

One can verify that the last inequality is equivalent to

$$
\sqrt{\rho} \leqslant \frac{2(1 + 3\sqrt{\eta})}{(1 + 3\sqrt{\eta})^2 + \left( \frac{1 + \sqrt{\eta}}{1 - \sqrt{\eta}} \right)^2},
$$

which always holds the assumption that $\rho \leqslant 0.18$ and $\eta \leqslant 0.01$.

Thus, we have proved the claimed lower bound on $\gamma_1$.

### C.1.1 A new Gaussian comparison inequality

We start with the following well-known comparison inequality, which was first derived in [17].

**Theorem 8** (Gordon's Gaussian comparison theorem)**.** Let $I, J \in \mathbb{N}^*$, and $\{X_{ij}\}, \{Y_{ij}\}$ be two centered Gaussian processed indexed on $I \times J$, such that for any $i, l \in I$ with $i \neq l$ and $j, k \in J$,

$$\begin{cases} \mathbb{E}X_{ij}^2 = \mathbb{E}Y_{ij}^2 \\ \mathbb{E}X_{ij}X_{ik} \geqslant \mathbb{E}Y_{ij}Y_{ik} \\ \mathbb{E}X_{ij}X_{lk} \leqslant \mathbb{E}Y_{ij}Y_{lk} \,. \end{cases}$$

Then, for any $\{\lambda_{ij}\} \in \mathbb{R}^{I \times J}$, we have

$$\mathbb{P}\left( \bigcap_{i=1}^{I} \bigcup_{j=1}^{J} [Y_{ij} \geqslant \lambda_{ij}] \right) \geqslant \mathbb{P}\left( \bigcap_{i=1}^{I} \bigcup_{j=1}^{J} [X_{ij} \geqslant \lambda_{ij}] \right)$$

Our next result is a consequence of Gordon's comparison inequality, and appears to be new. More specifically, it can be seen as a variant of the Sudakov-Fernique's inequality (see, for instance, Theorem 7.2.11 in [42]).

**Theorem 9.** Let $S_1 \subset \mathbb{R}^n$ and $S_2 \subset \mathbb{R}^m$ be non-empty sets, and $\psi : S_1 \times S_2 \to \mathbb{R}$ be a continuous function. Then, for any $c \in \mathbb{R}$,

$$\mathbb{P}\left( \sup_{(x,y)\in S_1 \times S_2} y^\top Gx + \psi(x,y) \geqslant c \right) \leqslant 2\,\mathbb{P}\left( \sup_{(x,y)\in S_1 \times S_2} \|x\|g^\top y + \|y\|h^\top x + \psi(x,y) \geqslant c \right),$$

*Proof.* The proof relies on several intermediate results, and is deferred to Section C.1.2. $\qquad\square$

**Lemma 4.** Let $G \in \mathbb{R}^{m \times n}$, $Z \in \mathbb{R}$, $g \in \mathbb{R}^m$ and $h \in \mathbb{R}^n$ have independent standard Gaussian entries. Let $I_1 \subset \mathbb{R}^n$ and $I_2 \subset \mathbb{R}^m$ be finite sets, and $\psi$ be a function defined over $I_1 \times I_2$. Then, for any $c \in \mathbb{R}$, we have

$$\mathbb{P}\left( \max_{(x,y)\in I_1 \times I_2} y^\top Gx + Z\|x\|\|y\| + \psi(x,y) \geqslant c \right) \leqslant \mathbb{P}\left( \max_{(x,y)\in I_1 \times I_2} \|x\|g^\top y + \|y\|h^\top x + \psi(x,y) \geqslant c \right).$$

*Proof.* We introduce two Gaussian processes $X$ and $Y$ indexed over $I_1 \times I_2$, defined as

$$X_{xy} = \|x\|g^\top y + \|y\|h^\top x, \qquad Y_{xy} = y^\top Gx + Z\|x\|\|y\|,$$

for all $(x,y) \in I_1 \times I_2$. It holds that $\mathbb{E}X_{xy} = \mathbb{E}Y_{xy} = 0$, $\mathbb{E}X_{xy}^2 = 2\|x\|^2\|y\|^2 = \mathbb{E}Y_{xy}^2$, and

$$\mathbb{E}[X_{xy}X_{x'y'}] = \|x\|\|x'\| y^\top y' + \|y\|\|y'\| x^\top x',$$
$$\mathbb{E}[Y_{xy}Y_{x'y'}] = \|x\| \|x'\| \|y\| \|y'\| + x^\top x' y^\top y'.$$

Consequently, we have

$$\mathbb{E}[Y_{xy}Y_{x'y'}] - \mathbb{E}[X_{xy}X_{x'y'}] = \left( \|x\| \|x'\| - x^\top x' \right) \left( \|y\| \|y'\| - y^\top y' \right)$$
$$\geqslant 0\,.$$

Therefore, applying Gordon's comparison theorem with $I = I_1 \times I_2$, $J$ being any finite set, and $\lambda_{xy} = \psi(x,y) - c$, we obtain that

$$\mathbb{P}\left( \min_{(x,y)\in I_1 \times I_2} y^\top Gx + Z\|x\|\|y\| - \psi(x,y) \geqslant -c \right) \geqslant \mathbb{P}\left( \min_{(x,y)\in I_1 \times I_2} \|x\|g^\top y + \|y\|h^\top x - \psi(x,y) \geqslant -c \right).$$

Using the symmetry of the Gaussian distribution, it follows that

$$\mathbb{P}\left( \max_{(x,y)\in I_1 \times I_2} y^\top Gx + Z\|x\|\|y\| + \psi(x,y) \leqslant c \right) \geqslant \mathbb{P}\left( \max_{(x,y)\in I_1 \times I_2} \|x\|g^\top y + \|y\|h^\top x + \psi(x,y) \leqslant c \right),$$

and consequently,

$$\mathbb{P}\left( \max_{(x,y)\in I_1 \times I_2} y^\top Gx + Z\|x\|\|y\| + \psi(x,y) \geqslant c \right) \leqslant \mathbb{P}\left( \max_{(x,y)\in I_1 \times I_2} \|x\|g^\top y + \|y\|h^\top x + \psi(x,y) \geqslant c \right),$$

$$\square$$

**Corollary 1.** Let $S_1 \subset \mathbb{R}^n$ and $S_2 \subset \mathbb{R}^m$ be non-empty sets, and $\psi : S_1 \times S_2 \to \mathbb{R}$ be a continuous function. Then, for any $c \in \mathbb{R}$,

$$\mathbb{P}\left( \sup_{(x,y) \in S_1 \times S_2} y^\top G x + Z\|x\|\|y\| + \psi(x,y) \geqslant c \right) \leqslant \mathbb{P}\left( \sup_{(x,y) \in S_1 \times S_2} \|x\|g^\top y + \|y\|h^\top x + \psi(x,y) \geqslant c \right),$$

*Proof.* According to Lemma 4, the result is true if $S_1$ and $S_2$ are finite. By monotone convergence, it is immediate to extend it to countable sets. By density arguments and monotone convergence, it also follows for any sets $S_1$ and $S_2$. □

### C.1.2 Proof of Theorem 9

We define $f_1(x,y) = y^\top G x + \psi(x,y)$ and $f_2(x,y) = y^\top G x + Z\|x\|\|y\| + \psi(x,y)$. If $Z > 0$, then $f_1 \leqslant f_2$ and $\sup_{x,y} f_1(x,y) \leqslant \sup_{x,y} f_2(x,y)$. Thus,

$$\mathbb{P}\left( \sup_{(x,y) \in S_1 \times S_2} f_1(x,y) \geqslant c, \quad Z > 0 \right) \leqslant \mathbb{P}\left( \sup_{(x,y) \in S_1 \times S_2} f_2(x,y) \geqslant c \right).$$

From Corollary 1, we know that

$$\mathbb{P}\left( \sup_{(x,y) \in S_1 \times S_2} f_2(x,y) \geqslant c \right) \leqslant \mathbb{P}\left( \sup_{(x,y) \in S_1 \times S_2} \|x\|g^\top y + \|y\|h^\top x + \psi(x,y) \geqslant c \right).$$

Consequently, using the independence of $f_1$ and $Z$, we get

$$\frac{1}{2}\mathbb{P}\left( \sup_{(x,y) \in S_1 \times S_2} f_1(x,y) \geqslant c \right) = \mathbb{P}\left( \sup_{(x,y) \in S_1 \times S_2} f_1(x,y) \geqslant c, \quad Z > 0 \right)$$

$$\leqslant \mathbb{P}\left( \sup_{(x,y) \in S_1 \times S_2} \|x\|g^\top y + \|y\|h^\top x + \psi(x,y) \geqslant c \right),$$

which yields the claim. □

## C.2 SRHT matrices – matrix deviation inequalities over ellipsoids

## C.3 Preliminaries

Let $S \in \mathbb{R}^{m \times n}$ be a SRHT matrix, that is, $S = RH\mathrm{diag}(\varepsilon)$ where $R$ is a row-subsampling matrix of size $m \times n$, $H$ is the normalized Walsh-Hadamard transform of size $n \times n$ and $\varepsilon$ is a vector of $n$ independent Rademacher variables. We introduce the scaled diagonal matrix $\bar{D} = \frac{D}{\|D\|_2}$. Note that $\|\bar{D}\|_F^2 = d_e$ and $\|\bar{D}\|_2 = 1$.

**Lemma 5.** Let $e_j$ be the $j$-th vector of the canonical basis in $\mathbb{R}^n$. Then,

$$\mathbb{P}\left\{ \max_{j=1,\dots,n} \|e_j^\top H\mathrm{diag}(\varepsilon)U\bar{D}\| \geqslant \sqrt{\frac{d_e}{n}} + \sqrt{\frac{8\log(\beta n)}{n}} \right\} \leqslant \frac{1}{\beta}. \tag{26}$$

*Proof.* We fix a row index $j \in \{1, \dots, n\}$, and define the function

$$f(x) := \|e_j^\top H\mathrm{diag}(x)U\bar{D}\| = \|x^\top EU\bar{D}\|,$$

where $E := \mathrm{diag}(e_j^\top H)$. Each entry of $E$ has magnitude $n^{-\frac{1}{2}}$. The function $f$ is convex, and its Lipschitz constant is upper bounded as follows,

$$|f(x) - f(y)| \leqslant \|(x-y)^\top EV\bar{D}\| \leqslant \|x-y\|\,\|E\|_2\,\|V\|_2\,\|\bar{D}\|_2 = \frac{1}{\sqrt{n}}\|x-y\|.$$

For a Rademacher vector $\varepsilon$, we have

$$\mathbb{E}f(\varepsilon) \leqslant \sqrt{\mathbb{E}f(\varepsilon)^2} = \|EU\bar{D}\|_F \leqslant \|EU\|_2\,\|\bar{D}\|_F = \sqrt{\frac{d_e}{n}}.$$

Applying Lipschitz concentration results for Rademacher variables, we obtain

$$\mathbb{P}\left\{\|e_j^\top H \mathrm{diag}(\varepsilon)U\bar{D}\| \geqslant \sqrt{\frac{d_e}{n}} + \sqrt{\frac{8\log(\beta n)}{n}}\right\} \leqslant \frac{1}{n\beta}.$$

Finally, taking a union bound over $j \in \{1,\dots,n\}$, we obtain the claimed result. $\qquad\square$

**Theorem 10** (Matrix Bernstein). *Let $\mathcal{X} = \{X_1,\dots,X_n\}$ be a finite set of squared matrices with dimension $d$. Fix a dimension $m$, and suppose that there exists a positive semi-definite matrix $V$ and a real number $K > 0$ such that $\mathbb{E}[X_I] = 0$, $\mathbb{E}[X_I^2] \preceq V$, and $\|X_I\|_2 \leqslant K$ almost surely, where $I$ is a uniformly random index over $\{1,\dots,n\}$. Let $T$ be a subset of $\{1,\dots,n\}$ with $m$ indices drawn uniformly at random without replacement. Then, for any $t \geqslant \sqrt{m\|V\|_2} + K/3$, we have*

$$\mathbb{P}\left\{\Big\|\sum_{i\in T}X_i\Big\|_2 \geqslant t\right\} \leqslant 8 \cdot d_e \cdot \exp\left(-\frac{t^2/2}{m\|V\|_2 + Kt/3}\right),$$

*where $d_e := \mathrm{tr}(V)/\|V\|$ is the intrinsic dimension of the matrix $V$.*

*Proof.* We denote $S_T := \sum_{i\in T}X_i$. Fix $\theta > 0$, define $\psi(t) = e^{\theta t} - \theta t - 1$, and use the Laplace matrix transform method (e.g., Proposition 7.4.1 in [41]) to obtain

$$\mathbb{P}\{\lambda_{\max}(S_T) \geqslant t\} \leqslant \frac{1}{\psi(t)}\mathbb{E}\,\mathrm{tr}\,\psi(S_T)$$

$$= \frac{1}{e^{\theta t} - \theta t - 1}\mathbb{E}\,\mathrm{tr}\left(e^{\theta S_T} - I\right),$$

and the last equality holds due to the fact that $\mathbb{E}\,S_T = m\,\mathbb{E}\,X_I = 0$. Let $T' = \{i_1,\dots,i_m\}$ be a subset of $\{1,\dots,n\}$, drawn uniformly at random *with* replacement. In particular, the indices of $T'$ are independent random variables, and so are the matrices $\{X_{i_j}\}_{j=1}^m$. Write $S_{T'} := \sum_{j=1}^m X_{i_j}$. Gross and Nesme [19] have shown that for any $\theta > 0$,

$$\mathbb{E}\,\mathrm{tr}\exp\left(\theta S_T\right) \leqslant \mathbb{E}\,\mathrm{tr}\exp\left(\theta S_{T'}\right).$$

As a consequence of Lieb's inequality (e.g., Lemma 3.4 in [41]), it holds that

$$\mathbb{E}\,\mathrm{tr}\exp\left(\theta S_{T'}\right) \leqslant \mathrm{tr}\exp\left(\sum_{j=1}^m \log\mathbb{E}\,e^{\theta X_{i_j}}\right) = \mathrm{tr}\exp\left(m\,\log\mathbb{E}\,e^{\theta X_I}\right).$$

Thus, it remains to bound $\mathbb{E}\,e^{\theta X_I}$. By assumption, $\mathbb{E}[X_I] = 0$ and $\|X_I\|_2 \leqslant K$ almost surely. Then, using Lemma 5.4.10 from [42], we get $\mathbb{E}\,e^{\theta X_I} \preceq \exp\left(g(\theta)\,\mathbb{E}\,X_I^2\right)$, for any $|\theta| < 3/K$ and where $g(\theta) = \frac{\theta^2/2}{1 - |\theta|K/3}$. By monotonicity of the logarithm, $m\cdot\log\mathbb{E}\,e^{\theta X_I} \preceq m\cdot g(\theta)\,\mathbb{E}\,X_I^2$. By assumption, $\mathbb{E}\,X_I^2 \preceq V$ and thus, $m\cdot\log\mathbb{E}\,e^{\theta X_I} \preceq m\cdot g(\theta)\,V$. By monotonicity of the trace exponential, it follows that $\mathrm{tr}\exp\left(m\,\log\mathbb{E}\,e^{\theta X_I}\right) \leqslant \mathrm{tr}\exp\left(m\,g(\theta)\,V\right)$, and further,

$$\mathbb{P}\{\lambda_{\max}(S_T) \geqslant t\} \leqslant \frac{1}{e^{\theta t} - \theta t - 1}\mathrm{tr}\left(e^{m\,g(\theta)\,V} - I\right) = \frac{1}{e^{\theta t} - \theta t - 1}\,\mathrm{tr}\,\varphi(m\,g(\theta)\,V),$$

where $\varphi(a) = e^a - 1$. The function $\varphi$ is convex, and the matrix $m\,g(\theta)\,V$ is positive semidefinite. Therefore, we can apply Lemma 7.5.1 from [41] and obtain

$$\mathrm{tr}\,\varphi(m\,g(\theta)\,V) \leqslant d_e\cdot\varphi(m\,g(\theta)\,\|V\|_2) \leqslant d_e\cdot e^{m\,g(\theta)\,\|V\|_2},$$

which further implies that

$$\mathbb{P}\{\lambda_{\max}(S_T) \geqslant t\} \leqslant d_e\cdot\frac{e^{\theta t}}{e^{\theta t} - \theta t - 1}\cdot e^{-\theta t + m\,g(\theta)\cdot\|V\|_2} \leqslant d_e\cdot\left(1 + \frac{3}{\theta^2 t^2}\right)\cdot e^{-\theta t + m\,g(\theta)\cdot\|V\|_2}.$$

For the last inequality, we used the fact that $\frac{e^a}{e^a - a - 1} = 1 + \frac{1+a}{e^a - a - 1} \leqslant 1 + \frac{3}{a^2}$ for all $a \geqslant 0$. Picking $\theta = t/(m\,\|V\|_2 + Kt/3)$, we obtain

$$\mathbb{P}\{\lambda_{\max}(S_T) \geqslant t\} \leqslant d_e\cdot\left(1 + 3\cdot\frac{(m\|V\|_2 + Kt/3)^2}{t^4}\right)\cdot\exp\left(-\frac{t^2/2}{m\|V\|_2 + Kt/3}\right).$$

Under the assumption $t \geqslant \sqrt{m\|V\|_2} + K/3$, the parenthesis in the above right-hand side is bounded by four, which results in

$$\mathbb{P}\left\{\lambda_{\max}(S_T) \geqslant t\right\} \leqslant 4 \cdot d_e \cdot \exp\left(-\frac{t^2/2}{m\|V\|_2 + Kt/3}\right).$$

Repeating the argument for $-S_T$ and combining the two bounds, we obtain the claimed result. $\quad\square$

### C.3.1 Proof of Theorem 4

We write $v_j := \sqrt{\frac{n}{m}}\, w_j$, where $w_j = e_j^\top H \mathrm{diag}(\varepsilon) U \bar{D}$, and $\varepsilon \in \{\pm 1\}^n$ is a fixed vector. We denote

$$\gamma := \max\left\{\max_{j=1,\ldots,n}\|v_j\|, m^{-\frac{1}{2}}\right\} \qquad \text{and} \qquad X_i := v_i v_i^\top - \frac{1}{m}\bar{D}^2.$$

Let $I$ be a uniformly random index over $\{1,\ldots,n\}$. We have

$$\mathbb{E}[X_I] = \frac{n}{m}\mathbb{E}[w_I w_I^\top] - \frac{1}{m}\bar{D}^2 = \frac{n}{m}\left(\frac{1}{n}\sum_{i=1}^n \bar{D} U^\top \mathrm{diag}(\varepsilon) H e_i e_i^\top H \mathrm{diag}(\varepsilon) U \bar{D}\right) - \frac{1}{m}\bar{D}^2$$

$$= \frac{1}{m}\bar{D} U^\top \mathrm{diag}(\varepsilon) H \underbrace{\sum_{i=1}^n e_i e_i^\top}_{=I} H \mathrm{diag}(\varepsilon) U \bar{D} - \frac{1}{m}\bar{D}^2$$

$$= 0.$$

The last equality holds due to the fact that $H^2 = I$, $\mathrm{diag}(\varepsilon)^2 = I$ and $U^\top U = I$. Further, $\|v_I\|^2 \leqslant \gamma^2$ a.s., so that $\|v_I\|^2 v_I v_I^\top \preceq \gamma^2 v_I v_I^\top$ a.s., and consequently, $\mathbb{E}\left[\|v_I\|^2 v_I v_I^\top\right] \preceq \gamma^2 \cdot \mathbb{E}[v_I v_I^\top]$. Thus,

$$\mathbb{E}\left[X_I^2\right] = \mathbb{E}\left[\|v_I\|^2 v_I v_I^\top\right] - \frac{2}{m}\bar{D}^4 + \frac{1}{m^2}\bar{D}^4$$

$$\leqslant \frac{\gamma^2}{m}\bar{D}^2 - \frac{2}{m^2}\bar{D}^4 + \frac{1}{m^2}\bar{D}^4$$

$$= \gamma^2 \cdot \frac{1}{m}\bar{D}^2 - \frac{1}{m^2}\bar{D}^4$$

$$\preceq \frac{\gamma^2}{m}\bar{D}^2.$$

The first inequality holds due to the fact that $\mathbb{E}\left[v_I v_I^\top\right] = m^{-1}\bar{D}^2$. Further, we have

$$\|X_I\| = \|v_I v_I^\top - \frac{1}{m}\bar{D}^2\| \leqslant \max\left\{\max_{j=1,\ldots,n}\|v_j\|^2, m^{-1}\right\} = \gamma^2.$$

Let $T$ be a subset of $m$ indices in $\{1,\ldots,n\}$ drawn uniformly at random, without replacement. Applying Theorem 10 with $V = m^{-1}\gamma^2 \bar{D}^2$ and using the scale invariance of the effective dimension, we obtain that for any $t \geqslant \gamma + \gamma^2/3$,

$$\mathbb{P}\left\{\left\|\sum_{i \in T} X_i\right\|_2 \geqslant t\right\} \leqslant 8 d_e \cdot \exp\left(-\frac{t^2/2}{\gamma^2(1 + t/3)}\right).$$

Suppose now that $\varepsilon$ is a vector of independent Rademacher variables. Note that $\sum_{i \in T} X_i \overset{\mathrm{d}}{=} \bar{D} U^\top (S^\top S - I) U \bar{D}$. From Lemma 5, we know that $\gamma \leqslant \sigma := \sqrt{\frac{d_e}{m}} + \sqrt{\frac{8 \log(d_e n)}{m}}$ with probability at least $1 - d_e^{-1}$. Consequently, with probability at least $1 - d_e^{-1} - 8 d_e \cdot \exp\left(-\frac{t^2/2}{\sigma^2(1+t/3)}\right)$, for $t \geqslant \sigma(1 + \sigma/3)$ we have

$$\left\|\bar{D} U^\top (S^\top S - I) U \bar{D}\right\|_2 \leqslant t. \tag{27}$$

We set $t = \sigma\sqrt{8/3\log d_e}$, and $\rho = \frac{d_e\log(d_e)C(n,d_e)}{m}$ where $C(n,d_e) = \frac{16}{3}\left(1 + \sqrt{\frac{8\log(d_e n)}{d_e}}\right)^2$. We choose $m$ large enough so that $\rho \leqslant \left(1 - (8/3\log d_e)^{-\frac{1}{2}}\right)^2$. Then, we get that

$$\mathbb{P}\left\{\left\|DU^\top(S^\top S - I)UD\right\|_2 \geqslant \|D\|_2^2 \cdot \sqrt{\rho}\right\} \leqslant \frac{9}{d_e},$$

which is the claimed result. $\qquad\square$

## D    Proofs of auxiliary results

### D.1    Proof of Lemma 1

Let $\{x_t\}$ be a sequence of iterates. Let $\overline{U}\,\overline{\Sigma}\,\overline{V}^\top$ be a singular value decomposition of $\overline{A}$. Denote $\overline{S} = \begin{bmatrix} S & 0 \\ 0 & I_d \end{bmatrix}$, so that $H_S = (SA)^\top SA + \nu^2 I_d = (\overline{S}\,\overline{A})^\top(\overline{S}\,\overline{A})$.

We have that $g_t = \overline{A}^\top\,\overline{A}(x_t - x^*)$ and thus,

$$
\begin{aligned}
g_t^\top H_S^{-1} g_t &= \langle \overline{A}^\top\,\overline{A}(x_t - x^*), (\overline{A}^\top\overline{S}^\top\overline{S}\,\overline{A})^{-1}\overline{A}^\top\,\overline{A}(x_t - x^*)\rangle \\
&= \langle \overline{A}(x_t - x^*), \overline{A}(\overline{A}^\top\,\overline{S}^\top\,\overline{S}\,\overline{A})^{-1}\overline{A}^\top\,\overline{A}(x_t - x^*)\rangle \\
&= \langle \overline{A}(x_t - x^*), \overline{U}\,\overline{\Sigma}\,\overline{V}^\top(\overline{V}\,\overline{\Sigma}\,\overline{U}^\top\,\overline{S}^\top\,\overline{S}\,\overline{U}\,\overline{\Sigma}\,\overline{V}^\top)^{-1}\overline{V}\,\overline{\Sigma}\,\overline{U}^\top\,\overline{A}(x_t - x^*)\rangle \\
&= \langle \overline{A}(x_t - x^*), \overline{U}(\overline{U}^\top\,\overline{S}^\top\,\overline{S}\,\overline{U})^{-1}\overline{U}^\top\,\overline{A}(x_t - x^*)\rangle \\
&= \langle \overline{U}^\top\,\overline{A}(x_t - x^*), (\overline{U}^\top\,\overline{S}^\top\,\overline{S}\,\overline{U})^{-1}\overline{U}^\top\,\overline{A}(x_t - x^*)\rangle\,.
\end{aligned}
$$

Observing that $\overline{U}^\top\,\overline{S}^\top\,\overline{S}\,\overline{U} = C_S$, it follows that $\frac{1}{2}g_t^\top H_S^{-1}g_t = \frac{1}{2}\|C_S^{-\frac{1}{2}}\overline{U}^\top\,\overline{A}(x_t - x^*)\|^2 = r_t$, which concludes the proof. $\qquad\square$

### D.2    Proof of Lemma 2

Fix $\rho \leqslant 0.18$ and $\eta \leqslant 0.01$. Let $a \geqslant 1$ be some numerical constant, and assume that the event $\mathcal{E}_{\rho/a,\eta}$ holds. Then, we have that

$$\sqrt{c_{\mathrm{gd}}(\rho/a,\eta)} = \frac{2}{\sqrt{a}}\frac{\sqrt{\rho c_\eta}}{1 + \frac{\rho c_\eta}{a}}\,, \qquad \sqrt{\frac{\sigma_{\max}(C_S)}{\sigma_{\min}(C_S)}} \leqslant \frac{\sqrt{a} + \sqrt{\rho c_\eta}}{\sqrt{a} - \sqrt{\rho c_\eta}}\,.$$

Using that $\sqrt{\rho c_\eta} \leqslant \sqrt{0.18 \cdot 1.3^2} \leqslant 0.56$ and $\rho c_\eta \leqslant 0.31$, we obtain that

$$
\begin{aligned}
\sqrt{c_{\mathrm{gd}}(\rho/a,\eta)} \cdot \sqrt{\frac{\sigma_{\max}(C_S)}{\sigma_{\min}(C_S)}} &\leqslant \frac{1}{\sqrt{a}}\frac{1 + \rho c_\eta}{1 + \frac{\rho c_\eta}{a}}\frac{\sqrt{a} + \sqrt{\rho c_\eta}}{\sqrt{a} - \sqrt{\rho c_\eta}} \cdot \sqrt{c_{\mathrm{gd}}(\rho,\eta)} \\
&\leqslant \frac{1.31}{\sqrt{a}} \cdot \frac{\sqrt{a} + 0.56}{\sqrt{a} - 0.56} \cdot \sqrt{c_{\mathrm{gd}}(\rho,\eta)}\,.
\end{aligned}
$$

The function $g : x \mapsto \frac{1.31}{\sqrt{x}} \cdot \frac{\sqrt{x}+0.56}{\sqrt{x}-0.56}$ is decreasing on $(0.56^2, +\infty)$ and $g(5) \leqslant 1$. Thus, for any $a \geqslant 5$, it holds that

$$c_{\mathrm{gd}}(\rho/a,\eta) \cdot \frac{\sigma_{\max}(C_S)}{\sigma_{\min}(C_S)} \leqslant c_{\mathrm{gd}}(\rho,\eta)\,,$$

and this concludes the proof. $\qquad\square$

### D.3    Proof of Lemma 3

By definition, we have on the event $\mathcal{E}_{\rho/a_\rho}$ that

$$\lambda_{\rho/a_\rho} \leqslant \sigma_{\min}(C_S) \leqslant \sigma_{\max}(C_S) \leqslant \Lambda_{\rho/a_\rho}\,,$$

where $\lambda_{\rho/a_\rho} = 1 - \sqrt{\frac{\rho}{a_\rho}}$ and $\Lambda_{\rho/a_\rho} = 1 + \sqrt{\frac{\rho}{a_\rho}}$, and $a_\rho = \frac{1+\sqrt{\rho}}{1-\sqrt{\rho}}$. It follows that

$$\frac{\sigma_{\max}(C_S)}{\sigma_{\min}(C_S)} \leqslant \frac{1 + \sqrt{\frac{\rho}{a_\rho}}}{1 - \sqrt{\frac{\rho}{a_\rho}}} = \frac{\sqrt{a_\rho} + \sqrt{\rho}}{\sqrt{a_\rho} - \sqrt{\rho}}.$$

The function $x \mapsto \frac{x+\sqrt{\rho}}{x-\sqrt{\rho}}$ is decreasing on $[1, +\infty)$. Since $a_\rho > 1$, it follows that $f(a_\rho) < f(1)$, i.e., $\frac{\sqrt{a_\rho}+\sqrt{\rho}}{\sqrt{a_\rho}-\sqrt{\rho}} < \frac{1+\sqrt{\rho}}{1-\sqrt{\rho}}$, i.e., $\frac{\sqrt{a_\rho}+\sqrt{\rho}}{\sqrt{a_\rho}-\sqrt{\rho}} < a_\rho$, which yields that

$$\frac{\sigma_{\max}(C_S)}{\sigma_{\min}(C_S)} \leqslant a_\rho.$$

Regarding the second statement of Lemma 3, a simple calculation yields that $c_{\mathrm{gd}}(\rho') = \rho'$ for any $\rho' \in (0,1)$. This further implies that $c_{\mathrm{gd}}(\rho/a_\rho) = \frac{\rho}{a_\rho} = \frac{c_{\mathrm{gd}}(\rho)}{a_\rho}$, which concludes the proof. $\qquad\square$

## Footnotes

[4]The spectral radius of a complex-valued matrix is the largest module of its complex eigenvalues.