[Reviews · NeurIPS 2020]

Review 1

Summary and Contributions: This paper proposes an iterative algorithm based on randomized sketching, which approximately solves the ridge regression problem in a faster way. The authors provide accuracy guarantees under SRHT or JL type sketching matrices.

Strengths: I like the idea of choosing the sketching dimension adaptively. This ends up with having a sketch size close to the effective dimension even without a prior knowledge of the latter.

Weaknesses: 1. The authors claim that this is the first algorithm where the sketch-size depends only on the effective dimension of the problem, and not on the actual dimensions of the data matrix. If I'm not missing anything, this is not true in general. For example, the following work presented an iterative algorithm for ridge regression where the sketch size depends only on the effective dimension: An iterative, sketching-based framework for ridge regression by Chowdhury, Yang, and Drineas (2018). Even though the aforementioned paper deals only with the underdetermined case, I think the guarantee for an overdetermined A can be trivially found using some standard iterative refinement (such as Richardson or CG) with a randomized (Blendenpik type) preconditioner. The accuracy bound will be slightly different from that of Theorem 7 of the current work. 2. The algorithm proposed in this paper doesn't exploit the sparsity of the data, whereas the running time of the algorithm in the aforementioned paper depends on the number of non-zero elements of A, as the choice of the sketching matrix S there is a combination of various standard sketches discussed in the following paper: Optimal approximate matrix product in terms of stable rank by Cohen, Nelson, and Woodruff (2015).

Correctness: The theoretical method as well as the empirical evaluations look correct.

Clarity: The paper is well written.

Relation to Prior Work: The authors discussed some of the related works.

Reproducibility: Yes

Additional Feedback: ===================== After the rebuttal ============================ Thanks for addressing my concerns. I have read other reviews and the authors feedback. Note that the paper I mentioned also has the sketch size m \approx d_e when S is constructed using the methods of Cohen, Nelson, and Woodruff (2015). On the other hand, when S is constructed using Algorithm 2 of the paper, then, as the authors mentioned the sketch-size is given by m \approx d_e log d_e. However, the uniqueness of the current paper indeed lies in adaptively adjusting the sketch-size without having a prior knowledge on d_e. Therefore, I increase the the score to accept. Secondly, what I didn't mention in the review is the quality of the output x_T in terms of the total variability. In other words, whether the mean-squared error (MSE) of x_T is close to the MSE of the optimal solution x^*. Along with sparse sketching matrices, it will also be great if the authors could throw some light on this in the revised version, may be as a future direction. ==============================================================


Review 2

Summary and Contributions: Authors provide an adaptive algorithm, based on Iterative Hessian Sketch, to solve the over-determined least-squares problem. This algorithm starts with an initial sketch size m=1, which is doubled when neither the gradient nor the Polyak update lead to sufficient convergence rates. This increase reaches a maximum which is related to the effective dimension of the problem d_e. This paper starts by proving concentrations inequalities for Gaussian and SRHT embeddings. Then, these bounds are used to prove the convergence of their adaptive method with high probability, especially, they explain that the number of skipped iterations K is finite and cannot be above a threshold. Finally, authors show empiricaly that this algorithm is competitive against conjugate gradient (CG) and preconditioned conjugate gradient (pre-CG) on real (MNIST and CIFAR10) and synthetic datasets.

Strengths: The authors provide an adaptive algorithm which incrementally increases the sketch size. Their method gives a better complexity, by a factor log(d_e) by dealing implicitly with embeddings related to d_e. This algorithm would allow practitioners to benefit from cases where d_e << d, where d is the initial dimension size, and achieve faster convergence than CG which performs well when the condition number is near 1, or pre-CG which involves a costly initial preconditioning step. They also give high probability convergence proofs in Theorem 5 & 6. Through their analysis, authors also provide estimates of bounds of the eigenvalues of Gaussian and SRHT embeddings which are sharper then previously known bounds when m is large.

Weaknesses: The algorithm seems to present several caveats: 1) performing GD or Polyak steps only if the improvement ratio is below given threshold is wasting a lot of gradient computations 2) the benefit of switching between GD and Polyak step is unclear (even if trying a Polyak step is not costly) 3) there are a lot of hyperparameters to give as input, i,e, the convergence rates, the step sizes and the momentum parameter. All this parameters are functions of the bounds of the eigenvalues of the matrix C_S. Authors state that in practice one should use estimates given in Def 3.1 and 3.2, yet they depend on two other parameters \rho and \eta. 4) even if the number of rejected updates K is finite, it might be very large if \eta gets close to 1 or \rho to 0

Correctness: Proofs are clear. The numerical experiments section is maybe not detailed enough.

Clarity: The paper is very clear, except few minor details: 5) maybe, in Theorem 1 & 2, the step size should be indexed by \mu_{gd} or \mu_{p} 6) Def 3.1 and Def 3.2 might have been splited in Definition + Lemma 7) There are two different notations for vector Euclidean norm 8) In the experiments question, it could be clearer if the analyses were outside of the caption of the Figures 9) In Figure 3, are the error bars so small that they are not possible to see ? 10) In proof C.1.2, two different object are called \Lambda : the diagonal matrix and the upper bound of the eigenvalues 11) How does a(\rho, \eta) appears in the bounds when applying the Theorem 3 in proof of Theorem 5. Maybe a step is skipped

Relation to Prior Work: Yes, the novelty of the adaptive algorithm is clear. Effectiveness of IHS and its complexity if well explained.

Reproducibility: No

Additional Feedback: 12) lines 118-120 : instead of comparing the number of iteration, comparing the total complexity may be here more precise 13) What are the precise lemma or remarks in the references which give the suggested values for m for Gaussian and SRHT embeddings Concerning the numerical experiments: 14) is the code available online ? 15) line 269: "less memory space". Is it measured on the RAM or theoretical ? 16) How are \rho and \eta chosen ? 17) Can you please give the precise reference or explanation why $m = d / \rho$ or $m = d \log(d) / \rho$ for p-CG ? 18) Is the preconditioning done from scratch for each value of the penalty \nu ? This would explain part of the slowness of p-CG. 19) What did you observe for \nu = 0 (ie without regularization) concerning the Polyak and GD steps? One would prefer that it applies only GD steps since momentum does not improve the performance of IHS. **************************************************************************************** I update here my review after having paid a lot of attention to the authors' feedback and the reviews of the other reviewers. I was convinced by the rave reviews of R3 and R4 and by your answers to main concerns in the feedback to upgrade your score.


Review 3

Summary and Contributions: The paper contains three major contributions in my opinion. First the authors show an upper bound for the required dimensionality of sketch in iterative Hessian sketching methods applied to regularized least squares regression problems that depends on the effective (statistical) dimensionality of problem instead of the number of variables; effective dimension can be significantly lower. [Similar results with effective dimensionality were known earlier only for matrix multiplication, or sketch and solve type methods to the best of my knowledge.] Second, since the effective dimension is not known upfront, they provide a theoretical and a heuristic improvement criteria, that lets them start with sketch size of 1, monitor the progress of the iterative method, and if the error reduction is insufficient, then double the size of the sketch. This way the size of the sketch adapts to the optimal size, which depends on the unknown effective dimension. Lastly, thorough empirical evaluation with MNIST, CIFAR, and synthetic data proves that their method regularly outperforms (randomized) preconditioned conjugate gradient descent, a SOTA baseline.

Strengths: - Least squares regression (and related problems) are broadly used, it's nice to see improvements to the toolbox used for solving these type of problems. - Thorough and precise theoretical analysis, the proofs rely on fairly sophisticated tools from probability theory. - The resulting algorithm is practical. - Strong experimental evaluation supports the theoretical claims. - The results also extend to the under-constrained (minimum norm) case by duality.

Weaknesses: - Improvement criteria C_t that is actually implementable, is chosen by modifying theoretically justified c_t. It seems to me that theoretical claims no longer hold for C_t (in the worst case, or they require more justification). Details: Line 208: "Provided that c_t and C_t are close enough, this would yield the desired performance." I'd like to see some more detailed analysis and arguments here. Could you argue why these would be close? When statistical dimension d_e is significantly below algebraic dimension d, then the rescaling matrix is far from diagonal. Also, c_t is independent of S, C_t depends on sketching matrix S, as it involves H_S^{-1}. Isn't it circular logic to measure the quality of S with ratios defined by S? E.g what if S strongly distorts a particular direction (it's far from isometry), yet that cancels in the ratio? Or could you pick some other overall error measure than sub optimality gap \delta_t such that the analysis goes through with C_t? - Code is not available yet. Could you please release it?

Correctness: The theoretical claims are justified and the experimental evaluation seems sound.

Clarity: The paper is well written and generally readable.

Relation to Prior Work: The paper does a fine job in positioning its results and citing related work.

Reproducibility: Yes

Additional Feedback: Formulas for d_e in Line 63 and Line 83 are different. The second has denominator ||D||_2^2, note that ||D||_2 < 1, and hence it's larger. Related work is based on the first version usually. Could you clarify whether you need a slightly larger and non-standard effective dim definition for the proofs to go through? Line 81: Use of \bar{A} precedes definition of \bar{A} in line 84. Line 117, minor historical note: result cited from [2], which follows from elementary linear algebra, was also stated earlier on slide 25 of the following talk in 2007: http://www.ipam.ucla.edu/programs/workshops/workshops-ii-numerical-tools-and-fast-algorithms-for-massive-data-mining-search-engines-and-applications/?tab=schedule Tamas Sarlos (Yahoo! Research) Faster Least Squares Approximation http://helper.ipam.ucla.edu/publications/sews2/sews2_7133.ppt Algorithm 1: Could you compare experimentally whether it's really advantageous to run Gradient-IHS and Polyak-IHS in parallel, which can increase flops (even if lines 4-6 are merged into branches of if statement)? Experiments: Could you compare with IHS with fixed sketching dimension chosen heuristically and in hindsight? Also, it would be nice to see how the sketch size evolves during iterations. How did you pick target improvement rates c_*? In some sense both target improvement rates and sketch sizes are hyper-parameters. If sketch sizes ramp up to final quickly and ideal c_* varies broadly across data sets, then choosing hyper-parameters of proposed method is not (much) easier than choosing a fixed sketch size for a simpler method. Can we have a graph that compares best method with Gaussian and best method with SRHT? I.e should one choose Gaussian or SRHT sketches? Section E.2 Similar theorems are already known when d_e is replaced with d. It's great to have their full proof for readability. Nevertheless could you please provide reference for the proof that resembles your proof the most, and highlight where and how you deviate to obtain these tighter bounds. Lemma 5: Would this follow from and equation (4) of https://www.cs.princeton.edu/~chazelle/pubs/FJLT-sicomp09.pdf and sub-multiplicativity of spectral norm? Theorem 10 Matrix Bernstein, see Theorem 1.4 in https://arxiv.org/abs/1004.4389


Review 4

Summary and Contributions: The authors consider the problem of sketched ridge regression, using iterative Hessian sketching (IHS) algorithms. In particular, they address the issue of choosing the sketching dimension adaptively in order to accelerate the convergence rate. The contributions are two-fold: (1) for Gaussian and SRHT sketching, they give rates of convergence of two IHS algorithms, and (2) they give an algorithm that uses empirical estimates of these convergence rates to select the sketching rates. The latter algorithm chooses between two variants of IHS updates at each time, and is shown to have a faster running time than previous randomly preconditioned ridge regression algorithms.

Strengths: I find the work compelling. On the theoretical side, previous randomly preconditioned ridge regression algorithms require accurate estimates of the effective dimension to avoid using over large sketches. This work gives an iterative algorithm where the sketch size is tuned over time to be on the order of the effective dimension, and the algorithm provably doesn't spend too much time in the tuning process. This is of great practical interest for large problems, as it allows effective memory use. The empirical evaluation demonstrates this point. I believe this work is of relevance to anyone in the NeurIPS community solving l2 regression problems, or using sketching as a tool.

Weaknesses: I did not notice any weaknesses in the stated theory, caveat being I did not have time to read the supplementary material. I would like to see an explanation of how the step-sizes were chosen in the empirical evaluations. Update: I have read the author's response, and my score remains accept.

Correctness: The claims and methods as presented in the main paper are plausible. The empirical methodology for comparing the proposed algorithm to the baselines (CG and randomly preconditioned CG) were also satisfactory.

Clarity: The ideas of the paper are clearly presented, but I would have preferred more simplified presentations of the Theorems in the main body of the paper. There are a few points where the authors say results follow from the stated Theorems that are not clear (e.g. line 157).

Relation to Prior Work: Yes

Reproducibility: No

Additional Feedback: Some minor issues: - on line 62, regularized is misspelt - on line 83, the definition of d_e should be ||D||_F^2, to match that used earlier on line 63 - in Figure 2, two of the figures should be labeled Gaussian

[Author Response · NeurIPS 2020]

We thank the reviewers for their careful reading and constructive comments. We will include their suggestions in the
final version, and we will release our code. In the remainder, we want to address the main points raised in the reviews.

*"It is not true that this is the first algorithm where the sketch-size depends only on the effective dimension [reference]."*
Thank you for pointing us this reference. We will include a detailed comparison. However, we believe that our work
provides the first sketching algorithm with sketch size no larger than the effective dimension and *without a priori*
*knowledge or estimation of the latter*. In the reference, Algorithm 2 takes as input a sketch size $m \approx d_e \log d_e$, which
requires estimation of $d_e$. The latter can be done efficiently, but in the restricted setting $d_e \leq (n+d)^{\frac{1}{3}}/\mathrm{poly}(\log(n+d))$
(please see, for instance, Theorem 60 in [2]). We believe that our method is more simple, and, has the advantage of
starting with an arbitrarily small sketch size. As illustrated in Figure 1, the sketch size can also remain smaller than
$\mathcal{O}(d_e/\rho)$. Lastly, our method applies to both the overdetermined and undetermined cases (please see Appendix B for
the latter).

*"The guarantee [of Theorem 7] can be trivially found [using CG with a randomized preconditioner]."* We emphasize
that our key contribution is to propose an adaptive sketch size algorithm. Using the IHS, we are able to monitor the
progress of our algorithm and adapt the sketch size accordingly. Extending such ideas to different methods based on
preconditioning + iterative refinement is an open question beyond the scope of our work.

*"The algorithm [...] doesn't exploit the sparsity of the data."* Although we aim to address the case of dense data matrices
which is standard in the literature, our method can be extended to sparse embeddings, for which similar concentration
bounds exist in the literature. We will include these additional results in the final version.

*"Many hyperparameters.", "Choice of step sizes?", "Choice of target improvement rate $c^*$?", "K might be very large if*
*$\eta$ gets close to 1 or $\rho$ to 0"* We emphasize that $\rho$ and $\eta$ are the only users' choice parameters, which will be clarified in
the revision with suggested values. Other hyperparameters are chosen as in Theorem 5, and specified by the values of $\rho$
and $\eta$. One should choose a small $\eta$ for the concentration bounds to be tighter, and a typical value which preserves a
small failure probability is $\eta = o(1/\sqrt{m})$. If one picks a small $\rho$ to get a fast convergence rate, then, to avoid many
rejection steps (which cannot exceed $\mathcal{O}(\log(d_e/\rho))$ according to Theorem 5), one can either choose a larger initial
value of $m$, or, multiply $m$ by a constant larger than 2 at each rejection. In numerical experiments, we chose $\rho = 0.2$
for MNIST and $\rho = 0.5$ for CIFAR10, and $\eta = 0.05$ which results in at most 5 sketch size rejections.

*"[Unclear] benefits of switching between GD and Polyak steps", "Is it really advantageous to run Gradient-IHS*
*and Polyak-IHS in parallel?", "Can we have a graph that compares the best methods?".* We emphasize that the
Polyak-IHS update is guaranteed to perform at least as well as the gradient-IHS update, at the expense of just one
gradient computation but theoretically guaranteed convergence. We have carried out additional numerical comparisons
of the time and flops of running both in parallel, for both Gaussian and SRHT embeddings. In a nutshell, we typically
observe that either most Polyak steps are accepted, or, most of them are rejected, and this holds for both embeddings.
Based on all our numerical evaluations, the hybrid method with SRHT embeddings is most often the most efficient both
in terms of time and memory (RAM) usage as well as provably convergent with the specified sharp rate.

*"How does the sketch size evolve during iterations?", "Could you compare with IHS with fixed sketching dimension*
*chosen heuristically and in hindsight?"* Figures 1.(e-h) show how the adaptive sketch size changes throughout the
algorithm. Importantly, it can remain much smaller than the effective dimension. Without adaptation, for small sketch
sizes, the IHS fails to converge. We will include a detailed numerical comparison.

*"Is the preconditioning [of pCG] done from scratch for each value of the penalty $\nu$?"* We leverage previous iterates for
the preconditioning as we move along the regularization path.

*"In Figure 3, [are there] error bars?"* Error bars are reported on Figures 3(a–d). We will make them more readable.
*"How does $a(\rho, \eta)$ appears in the bounds when applying Theorem 3 in the proof of Theorem 5."* In the proof of Theorem
5, we apply Theorem 3 with $m$ greater than $d_e \, a(\rho, \eta)/\rho$. We will make these details clearer.

*"It seems to me that theoretical claims no longer hold for [the improvement ratio] $C_t$."* Our algorithm is guaranteed
to converge by monitoring the ratio $c_t$, although there is indeed a gap between the $C_t$ and $c_t$, which is controlled by
the condition number of the matrix $C_S$. Consequently, we pay an additional factor $(1 + \sigma_1^2/\nu^2)$ in the convergence
guarantee. Please see the proof of Theorem 5 for more details.

*"Formulas for $d_e$ in lines 63 and 83 are different"* We define formally $d_e$ on line 83. We will clarify this.

*"Similar theorems are already known when $d_e$ is replaced with $d$."* We will highlight differences between our analysis
techniques which result in sharper bounds and existing ones.

*"Precise reference for [...] $m = d \log(d)/\rho$ for p-CG ?"* Given a target convergence rate $\rho$, Lemma 1 in [24] specifies
$m = d^2/\rho$ for the SRHT. Tighter (and more recent) concentration bounds on the SRHT (see Lemma 3.4 [26]) suggest
to use $m = d \log d/\rho$.

[Meta-Review · NeurIPS 2020]

The paper presents a version of the Iterative Hessian Sketch, a well-known sketching method that solves problems such as ridge regression, which is adaptive, in the sense that it automatically adjusts the number of samples to take. This is important, since an incorrect guess on the number of samples can lead the error to increase exponentially, rather than decay exponentially. Besides having a nice result, the algorithm appears to work well in practice too, with the authors showing empirically that this algorithm is competitive against conjugate gradient (CG) and preconditioned conjugate gradient (pre-CG) on real (MNIST and CIFAR10) and synthetic datasets The rebuttal answered many of the issues of R1. In the end, all reviewers had very positive scores, and I have a lot of confidence in these reviewers. This is a clear accept.